# On Learning Over-parameterized Neural Networks: A Functional Approximation Perspective

**Lili Su**
CSAIL, MIT
lilisu@mit.edu

Pengkun Yang
Department of Electrical Engineering
Princeton University
pengkuny@princeton.edu

## Abstract

We consider training over-parameterized two-layer neural networks with Rectified Linear Unit (ReLU) using gradient descent (GD) method. Inspired by a recent line of work, we study the evolutions of network prediction errors across GD iterations, which can be neatly described in a matrix form. When the network is sufficiently over-parameterized, these matrices individually approximate *an* integral operator which is determined by the feature vector distribution $\rho$ only. Consequently, GD method can be viewed as *approximately* applying the powers of this integral operator on the underlying function $f^*$ that generates the responses. We show that if $f^*$ admits a low-rank approximation with respect to the eigenspaces of this integral operator, then the empirical risk decreases to this low-rank approximation error at a linear rate which is determined by $f^*$ and $\rho$ only, i.e., the rate is independent of the sample size $n$. Furthermore, if $f^*$ has zero low-rank approximation error, then, as long as the width of the neural network is $\Omega(n \log n)$, the empirical risk decreases to $\Theta(1/\sqrt{n})$. To the best of our knowledge, this is the first result showing the sufficiency of nearly-linear network over-parameterization. We provide an application of our general results to the setting where $\rho$ is the uniform distribution on the spheres and $f^*$ is a polynomial. Throughout this paper, we consider the scenario where the input dimension $d$ is fixed.

## 1  Introduction

Neural networks have been successfully applied in many real-world machine learning applications. However, a thorough understanding of the theory behind their practical success, even for two-layer neural networks, is still lacking. For example, despite learning optimal neural networks is provably NP-complete [BG17, BR89], in practice, even the neural networks found by the simple first-order methods perform well [KSH12]. Additionally, in sharp contrast to traditional learning theory, over-parameterized neural networks (more parameters than the size of the training dataset) are observed to enjoy smaller training and even smaller generalization errors [ZBH+16]. In this paper, we focus on training over-parameterized two-layer neural networks with Rectified Linear Unit (ReLU) using gradient descent (GD) method. Our results can be extended to other activation functions that satisfy some regularity conditions; see [GMMM19, Theorem 2] for an example. The techniques derived and insights obtained in this paper might be applied to deep neural networks as well, for which similar matrix representation exists [DZPS18].

Significant progress has been made in understanding the role of over-parameterization in training neural networks with first-order methods [AZLL18, DZPS18, ADH+19, OS19, MMN18, LL18, ZCZG18, DLL+18, AZLS18, CG19]; with proper random network initialization, (stochastic) GD converges to a (nearly) global minimum provided that the width of the network $m$ is *polynomially* large in the size of the training dataset $n$. However, neural networks seem to interpolate the training

data as soon as the number of parameters exceed the size of the training dataset by a constant factor [ZBH$^+$16, OS19]. To the best of our knowledge, a provable justification of why such mild over-parametrization is sufficient for successful gradient-based training is still lacking. Moreover, the convergence rates derived in many existing work approach 0 as $n \to \infty$; see Section A in *Supplementary Material* for details. In many applications the volumes of the datasets are huge – the ImageNet dataset [DDS$^+$09] has 14 million images. For those applications, a non-diminishing (i.e., constant w. r. t. $n$) convergence rate is more desirable. In this paper, our goal is to characterize a *constant* (w. r. t. $n$) convergence rate while improving the sufficiency guarantee of network over-parameterization. Throughout this paper, we focus on the setting where the dimension of the feature vector $d$ is fixed, leaving the high dimensional region as one future direction.

Inspired by a recent line of work [DZPS18, ADH$^+$19], we focus on characterizing the evolutions of the neural network prediction errors under GD method. This focus is motivated by the fact that the neural network representation/approximation of a given function might not be unique [KB18], and this focus is also validated by experimental neuroscience [MG06, ASCC18].

**Contributions** It turns out that the evolution of the network prediction error can be neatly described in a matrix form. When the network is sufficiently over-parameterized, the matrices involved individually approximate *an* integral operator which is determined by the feature vector distribution $\rho$ only. Consequently, GD method can be viewed as *approximately* applying the powers of this integral operator on the underlying/target function $f^*$ that generates the responses/labels. The advantages of taking such a functional approximation perspective are three-fold:

- We showed in Theorem 2 and Corollary 1 that the existing rate characterizations in the influential line of work [DZPS18, ADH$^+$19, DLL$^+$18] approach zero (i.e., $\to 0$) as $n \to \infty$. This is because the spectra of these matrices, as $n$ diverges, concentrate on the spectrum of the integral operator, in which the unique limit of the eigenvalues is zero.

- We show in Theorem 4 that the training convergence rate is determined by how $f^*$ can be decomposed into the eigenspaces of an integral operator. This observation is also validated by a couple of empirical observations: (1) The spectrum of the MNIST data concentrates on the first a few eigenspaces [LBB$^+$98]; and (2) the training is slowed down if labels are partially corrupted [ZBH$^+$16, ADH$^+$19].

- We show in Corollary 2 that if $f^*$ can be decomposed into a finite number of eigenspaces of the integral operator, then $m = \Theta(n \log n)$ is sufficient for the training error to converge to $\Theta(1/\sqrt{n})$ with a constant convergence rate. To the best of our knowledge, this is the first result showing the sufficiency of nearly-linear network over-parameterization.

**Notations** For any $n, m \in \mathbb{N}$, let $[n] := \{1, \cdots, n\}$ and $[m] := \{1, \cdots, m\}$. For any $d \in \mathbb{N}$, denote the unit sphere as $\mathcal{S}^{d-1} := \{x : x \in \mathbb{R}^d, \& \|x\| = 1\}$, where $\|\cdot\|$ is the standard $\ell_2$ norm when it is applied to a vector. We also use $\|\cdot\|$ for the spectral norm when it is applied to a matrix. The Frobenius norm of a matrix is denoted by $\|\cdot\|_F$. Let $L^2(\mathcal{S}^{d-1}, \rho)$ denote the space of functions with finite norm, where the inner product $\langle \cdot, \cdot \rangle_\rho$ and $\| \cdot \|_\rho^2$ are defined as $\langle f, g \rangle_\rho := \int_{\mathcal{S}^{d-1}} f(x) g(x) \mathrm{d}\rho(x)$ and $\|f\|_\rho^2 := \int_{\mathcal{S}^{d-1}} f^2(x) \mathrm{d}\rho(x) < \infty$. We use standard Big-$O$ notations, e.g., for any sequences $\{a_r\}$ and $\{b_r\}$, we say $a_r = O(b_r)$ or $a_r \lesssim b_r$ if there is an absolute constant $c > 0$ such that $\frac{a_r}{b_r} \leq c$, we say $a_r = \Omega(b_r)$ or $a_r \gtrsim b_r$ if $b_r = O(a_r)$ and we say $a_r = \omega(b_r)$ if $\lim_{r \to \infty} |a_r/b_r| = \infty$.

## 2 Problem Setup and Preliminaries

**Statistical learning** We are given a training dataset $\{(x_i, y_i) : 1 \leq i \leq n\}$ which consists of $n$ tuples $(x_i, y_i)$, where $x_i$'s are feature vectors that are identically and independently generated from a common but *unknown* distribution $\rho$ on $\mathbb{R}^d$, and $y_i = f^*(x_i)$. We consider the problem of learning the unknown function $f^*$ with respect to the square loss. We refer to $f^*$ as a *target function*. For simplicity, we assume $x_i \in \mathcal{S}^{d-1}$ and $y_i \in [-1, 1]$. In this paper, we restrict ourselves to the family of $\rho$ that is absolutely continuous with respect to Lebesgue measure. We are interested in finding a neural network to approximate $f^*$. In particular, we focus on two-layer fully-connected neural

networks with ReLU activation, i.e.,

$$f_{\boldsymbol{W},\boldsymbol{a}}(x) = \frac{1}{\sqrt{m}} \sum_{j=1}^{m} a_j \left[ \langle x, w_j \rangle \right]_+, \quad \forall x \in \mathcal{S}^{d-1}, \tag{1}$$

where $m$ is the number of hidden neurons and is assumed to be even, $\boldsymbol{W} = (w_1, \cdots, w_m) \in \mathbb{R}^{d \times m}$ are the weight vectors in the first layer, $\boldsymbol{a} = (a_1, \cdots, a_m)$ with $a_j \in \{-1, 1\}$ are the weights in the second layer, and $[\cdot]_+ := \max\{\cdot, 0\}$ is the ReLU activation function.

Many authors assume $f^*$ is also a neural network [MMN18, AZLL18, SS96, LY17, Tia16]. Despite this popularity, a target function $f^*$ is not necessarily a neural network. One advantage of working with $f^*$ directly is, as can be seen later, certain properties of $f^*$ are closely related to whether $f^*$ can be learned quickly by GD method or not. Throughout this paper, for simplicity, we do not consider the scaling in $d$ and treat $d$ as a constant.

**Empirical risk minimization via gradient descent** For each $k = 1, \cdots, m/2$: Initialize $w_{2k-1} \sim \mathcal{N}(\boldsymbol{0}, \boldsymbol{I})$, and $a_{2k-1} = 1$ with probability $\frac{1}{2}$, and $a_{2k-1} = -1$ with probability $\frac{1}{2}$. Initialize $w_{2k} = w_{2k-1}$ and $a_{2k} = -a_{2k-1}$. All randomnesses in this initialization are independent, and are independent of the dataset. This initialization is chosen to guarantee zero output at initialization. Similar initialization is adopted in [CB18, Section 3] and [WGL+19]. [1] We fix the second layer $\boldsymbol{a}$ and optimize the first layer $\boldsymbol{W}$ through GD on the empirical risk w. r. t. square loss [2]:

$$L_n(\boldsymbol{W}) := \frac{1}{2n} \sum_{i=1}^{n} \left[ (y_i - f_{\boldsymbol{W}}(x_i))^2 \right]. \tag{2}$$

For notational convenience, we drop the subscript $\boldsymbol{a}$ in $f_{\boldsymbol{W},\boldsymbol{a}}$. The weight matrix $\boldsymbol{W}$ is update as

$$\boldsymbol{W}^{t+1} = \boldsymbol{W}^t - \eta \frac{\partial L_n(\boldsymbol{W}^t)}{\partial \boldsymbol{W}^t}, \tag{3}$$

where $\eta > 0$ is stepsize/learning rate, and $\boldsymbol{W}^t$ is the weight matrix at the end of iteration $t$ with $\boldsymbol{W}^0$ denoting the initial weight matrix. For ease of exposition, let

$$\widehat{y}_i(t) := f_{\boldsymbol{W}^t}(x_i) = \frac{1}{\sqrt{m}} \sum_{j=1}^{m} a_j \left[ \langle w_j^t, x_i \rangle \right]_+, \quad \forall i = 1, \cdots, n. \tag{4}$$

Notably, $\widehat{y}_i(0) = 0$ for $i = 1, \cdots, n$. It can be easily deduced from (3) that $w_j$ is updated as

$$w_j^{t+1} = w_j^t + \frac{\eta a_j}{n\sqrt{m}} \sum_{i=1}^{n} (y_i - \widehat{y}_i(t)) x_i \mathbf{1}_{\left\{ \langle w_j^t, x_i \rangle > 0 \right\}}. \tag{5}$$

**Matrix representation** Let $\boldsymbol{y} \in \mathbb{R}^n$ be the vector that stacks the responses of $\{(x_i, y_i)\}_{i=1}^{n}$. Let $\widehat{\boldsymbol{y}}(t)$ be the vector that stacks $\widehat{y}_i(t)$ for $i = 1, \cdots, n$ at iteration $t$. Additionally, let $\mathcal{A} := \{j : a_j = 1\}$ and $\mathcal{B} := \{j : a_j = -1\}$. The evolution of $(\boldsymbol{y} - \widehat{\boldsymbol{y}}(t))$ can be neatly described in a matrix form. Define matrices $\boldsymbol{H}^+, \widetilde{\boldsymbol{H}}^+$, and $\boldsymbol{H}^-, \widetilde{\boldsymbol{H}}^-$ in $\mathbb{R}^n \times \mathbb{R}^n$ as: For $t \geq 0$, and $i, i' \in [n]$,

$$\boldsymbol{H}_{ii'}^+(t+1) = \frac{1}{nm} \langle x_i, x_{i'} \rangle \sum_{j \in \mathcal{A}} \mathbf{1}_{\left\{ \langle w_j^t, x_{i'} \rangle > 0 \right\}} \mathbf{1}_{\left\{ \langle w_j^t, x_i \rangle > 0 \right\}}, \tag{6}$$

$$\widetilde{\boldsymbol{H}}_{ii'}^+(t+1) = \frac{1}{nm} \langle x_i, x_{i'} \rangle \sum_{j \in \mathcal{A}} \mathbf{1}_{\left\{ \langle w_j^t, x_{i'} \rangle > 0 \right\}} \mathbf{1}_{\left\{ \langle w_j^{t+1}, x_i \rangle > 0 \right\}}, \tag{7}$$

and $\boldsymbol{H}_{ii'}^-(t+1), \widetilde{\boldsymbol{H}}_{ii'}^-(t+1)$ are defined similarly by replacing the summation over all the hidden neurons in $\mathcal{A}$ in (6) and (7) by the summation over $\mathcal{B}$. It is easy to see that both $\boldsymbol{H}^+$ and $\boldsymbol{H}^-$ are

positive semi-definite. The only difference between $\boldsymbol{H}_{ii'}^+(t+1)$ (or $\boldsymbol{H}_{ii'}^-(t+1)$) and $\widetilde{\boldsymbol{H}}_{ii'}^+(t+1)$ (or $\widetilde{\boldsymbol{H}}_{ii'}^-(t+1)$) is that $\mathbf{1}_{\left\{\langle w_j^t, x_i\rangle > 0\right\}}$ is used in the former, whereas $\mathbf{1}_{\left\{\langle w_j^{t+1}, x_i\rangle > 0\right\}}$ is adopted in the latter. When a neural network is sufficiently over-parameterized (in particular, $m = \Omega(\text{poly}(n))$), the sign changes of the hidden neurons are sparse; see [AZLL18, Lemma 5.4] and [ADH$^+$19, Lemma C.2] for details. The sparsity in sign changes suggests that both $\widetilde{\boldsymbol{H}}^+(t) \approx \boldsymbol{H}^+(t)$ and $\widetilde{\boldsymbol{H}}^-(t) \approx \boldsymbol{H}^-(t)$ are approximately PSD.

**Theorem 1.** *For any iteration $t \geq 0$ and any stepsize $\eta > 0$, it is true that*

$$\left(\boldsymbol{I} - \eta\left(\widetilde{\boldsymbol{H}}^+(t+1) + \boldsymbol{H}^-(t+1)\right)\right)(\boldsymbol{y} - \widehat{\boldsymbol{y}}(t))$$
$$\leq (\boldsymbol{y} - \widehat{\boldsymbol{y}}(t+1))$$
$$\leq \left(\boldsymbol{I} - \eta\left(\boldsymbol{H}^+(t+1) + \widetilde{\boldsymbol{H}}^-(t+1)\right)\right)(\boldsymbol{y} - \widehat{\boldsymbol{y}}(t)),$$

*where the inequalities are entry-wise.*

Theorem 1 says that when the sign changes are sparse, the dynamics of $(\boldsymbol{y} - \widehat{\boldsymbol{y}}(t))$ are governed by a sequence of PSD matrices. Similar observation is made in [DZPS18, ADH$^+$19].

# 3 Main Results

We first show (in Section 3.1) that the existing convergence rates that are derived based on minimum eigenvalues approach 0 as the sample size $n$ grows. Then, towards a non-diminishing convergence rate, we characterize (in Section 3.2) how the target function $f^*$ affects the convergence rate.

## 3.1 Convergence rates based on minimum eigenvalues

Let $\boldsymbol{H} := \boldsymbol{H}^+(1) + \boldsymbol{H}^-(1)$. It has been shown in [DZPS18] that when the neural networks are sufficiently over-parameterized $m = \Omega(n^6)$, the convergence of $\|\boldsymbol{y} - \widehat{\boldsymbol{y}}(t)\|$ and the associated convergence rates with high probability can be upper bounded as [3]

$$\|\boldsymbol{y} - \widehat{\boldsymbol{y}}(t)\| \leq (1 - \eta\lambda_{\min}(\boldsymbol{H}))^t \|\boldsymbol{y} - \widehat{\boldsymbol{y}}(0)\| = \exp\left(-t\log\frac{1}{1 - \eta\lambda_{\min}(\boldsymbol{H})}\right)\|\boldsymbol{y}\|, \quad (8)$$

where $\lambda_{\min}(\boldsymbol{H})$ is the smallest eigenvalue of $\boldsymbol{H}$. Equality (8) holds because of $\widehat{\boldsymbol{y}}(0) = \boldsymbol{0}$. In this paper, we refer to $\log\frac{1}{1 - \eta\lambda_{\min}(\boldsymbol{H})}$ as *convergence rate*. The convergence rate here is quite appealing at first glance as it is *independent* of the target function $f^*$. Essentially (8) says that no matter how the training data is generated, via GD, we can always find an over-parameterized neural network that perfectly fits/memorizes all the training data tuples exponentially fast! Though the spectrum of the random matrix $\boldsymbol{H}$ can be proved to concentrate as $n$ grows, we observe that $\lambda_{\min}(\boldsymbol{H})$ converges to 0 as $n$ diverges, formally shown in Theorem 2.

**Theorem 2.** *For any data distribution $\rho$, there exists a sequence of non-negative real numbers $\lambda_1 \geq \lambda_2 \geq \ldots$ (independent of $n$) satisfying $\lim_{i\to\infty}\lambda_i = 0$ such that, with probability $1 - \delta$,*

$$\sup_i |\lambda_i - \widetilde{\lambda}_i| \leq \sqrt{\frac{\log(4n^2/\delta)}{m}} + \sqrt{\frac{8\log(4/\delta)}{n}}. \quad (9)$$

*where $\widetilde{\lambda}_1 \geq \cdots \geq \widetilde{\lambda}_n$ are the spectrum of $\boldsymbol{H}$. In addition, if $m = \omega(\log n)$, we have*

$$\lambda_{\min}(\boldsymbol{H}) \xrightarrow{\mathbb{P}} 0, \quad \text{as } n \to \infty, \quad (10)$$

*where $\xrightarrow{\mathbb{P}}$ denotes convergence in probability.*

A numerical illustration of the decay of $\lambda_{\min}(\boldsymbol{H})$ in $n$ is presented in Fig. 1a. Theorem 2 is proved in Appendix D. By Theorem 2, the convergence rate in (8) approaches zero as $n \to \infty$.

**Corollary 1.** *For any $\eta = O(1)$, it is true that $\log\frac{1}{1 - \eta\lambda_{\min}(\boldsymbol{H})} \to 0$ as $n \to \infty$.*

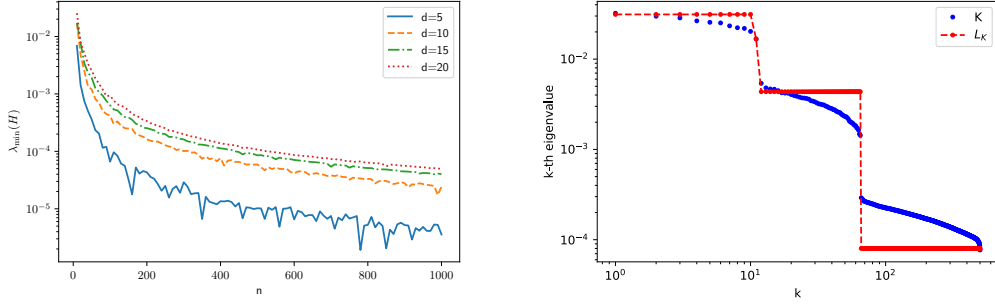

(a) The minimum eigenvalues of one realization of $\boldsymbol{H}$ under different $n$ and $d$, with network width $m = 2n$.

(b) The spectrum of $\boldsymbol{K}$ with $d = 10, n = 500$ concentrates around that of $L_{\mathcal{K}}$.

Figure 1: The spectra of $\boldsymbol{H}, \boldsymbol{K}$, and $L_{\mathcal{K}}$ when $\rho$ is the uniform distribution over $\mathcal{S}^{d-1}$.

In Corollary 1, we restrict our attention to $\eta = O(1)$. This is because the general analysis of GD [Nes18] adopted by [ADH+19, DZPS18] requires that $(1 - \eta\lambda_{\max}(\boldsymbol{H})) > 0$, and by the spectrum concentration given in Theorem 2, the largest eigenvalue of $\boldsymbol{H}$ concentrates on some strictly positive value as $n$ diverges, i.e., $\lambda_{\max}(\boldsymbol{H}) = \Theta(1)$. Thus, if $\eta = \omega(1)$, then $(1 - \eta\lambda_{\max}(\boldsymbol{H})) < 0$ for any sufficiently large $n$, violating the condition assumed in [ADH+19, DZPS18].

Theorem 2 essentially follows from two observations. Let $\boldsymbol{K} = \mathbb{E}[\boldsymbol{H}]$, where the expectation is taken with respect to the randomness in the network initialization. It is easy to see that by standard concentration argument, for a given dataset, the spectrum of $\boldsymbol{K}$ and $\boldsymbol{H}$ are close with high probability. In addition, the spectrum of $\boldsymbol{K}$, as $n$ increases, concentrates on the spectrum of the following integral operator $L_{\mathcal{K}}$ on $L^2(\mathcal{S}^{d-1}, \rho)$,

$$(L_{\mathcal{K}}f)(x) := \int_{\mathcal{S}^{d-1}} \mathcal{K}(x, s)f(s)\mathrm{d}\rho, \tag{11}$$

with the kernel function:

$$\mathcal{K}(x, s) := \frac{\langle x, s \rangle}{2\pi}\left(\pi - \arccos\langle x, s\rangle\right) \quad \forall x, s \in \mathcal{S}^{d-1}, \tag{12}$$

which is bounded over $\mathcal{S}^{d-1} \times \mathcal{S}^{d-1}$. In fact, $\lambda_1 \geq \lambda_2 \geq \cdots$ in Theorem 2 are the eigenvalues of $L_{\mathcal{K}}$. As $\sup_{x,s \in \mathcal{S}^{d-1}} \mathcal{K}(x, s) \leq \frac{1}{2}$, it is true that $\lambda_i \leq 1$ for all $i \geq 1$. Notably, by definition, $\boldsymbol{K}_{ii'} = \mathbb{E}[\boldsymbol{H}_{ii'}] = \frac{1}{n}\mathcal{K}(x_i, x_{i'})$ is the empirical kernel matrix on the feature vectors of the given dataset $\{(x_i, y_i) : 1 \leq i \leq n\}$. A numerical illustration of the spectrum concentration of $\boldsymbol{K}$ is given in Fig. 1b; see, also, [XLS17].

Though a generalization bound is given in [ADH+19, Theorem 5.1 and Corollary 5.2], it is unclear how this bound scales in $n$. In fact, if we do not care about the structure of the target function $f^*$ and allow $\frac{\boldsymbol{y}}{\sqrt{n}}$ to be arbitrary, this generalization bound might not decrease to zero as $n \to \infty$. A detailed argument and a numerical illustration can be found in Appendix B.

## 3.2 Constant convergence rates

Recall that $f^*$ denotes the underlying function that generates output labels/responses (i.e., $y$'s) given input features (i.e., $x$'s). For example, $f^*$ could be a constant function or a linear function. Clearly, the difficulty in learning $f^*$ via training neural networks should crucially depend on the properties of $f^*$ itself. We observe that the training convergence rate might be determined by how $f^*$ can be decomposed into the eigenspaces of the integral operator defined in (11). This observation is also validated by a couple of existing empirical observations: (1) The spectrum of the MNIST data [LBB+98] concentrates on the first a few eigenspaces; and (2) the training is slowed down if labels are partially corrupted [ZBH+16, ADH+19]. Compared with [ADH+19], we use spectral projection concentration to show how the random eigenvalues and the random projections in [ADH+19, Eq.(8) in Theorem 4.1] are controlled by $f^*$ and $\rho$.

We first present a sufficient condition for the convergence of $\|\boldsymbol{y} - \widehat{\boldsymbol{y}}(t)\|$.

**Theorem 3** (Sufficiency). *Let $0 < \eta < 1$. Suppose there exist $c_0 \in (0,1)$ and $c_1 > 0$ such that*

$$\left\| \frac{1}{\sqrt{n}} (\boldsymbol{I} - \eta \boldsymbol{K})^t \boldsymbol{y} \right\| \le (1 - \eta c_0)^t + c_1, \quad \forall\, t. \tag{13}$$

*For any $\delta \in (0, \frac{1}{4})$ and given $T > 0$, if*

$$m \ge \frac{32}{c_1^2} \left( \left( \frac{1}{c_0} + 2\eta T c_1 \right)^4 + 4 \log \frac{4n}{\delta} \left( \frac{1}{c_0} + 2\eta T c_1 \right)^2 \right), \tag{14}$$

*then with probability at least $1 - \delta$, the following holds for all $t \le T$:*

$$\left\| \frac{1}{\sqrt{n}} (\boldsymbol{y} - \widehat{\boldsymbol{y}}(t)) \right\| \le (1 - \eta c_0)^t + 2c_1. \tag{15}$$

Theorem 3 is proved in Appendix E. Theorem 3 says that if $\left\| \frac{1}{\sqrt{n}} (\boldsymbol{I} - \eta \boldsymbol{K})^t \boldsymbol{y} \right\|$ converges to $c_1$ exponentially fast, then $\left\| \frac{1}{\sqrt{n}} (\boldsymbol{y} - \widehat{\boldsymbol{y}}(t)) \right\|$ converges to $2c_1$ with the same convergence rate guarantee provided that the neural network is sufficiently parametrized. Recall that $y_i \in [-1, 1]$ for each $i \in [n]$. Roughly speaking, in our setup, $y_i = \Theta(1)$ and $\|\boldsymbol{y}\| = \sqrt{\sum_{i=1}^n y_i^2} = \Theta(\sqrt{n})$. Thus we have the $\frac{1}{\sqrt{n}}$ scaling in (13) and (14) for normalization purpose.

Similar results were shown in [DZPS18, ADH+19] with $\eta = \frac{\lambda_{\min}(\boldsymbol{K})}{n}$, $c_0 = n\lambda_{\min}(\boldsymbol{K})$ and $c_1 = 0$. But the obtained convergence rate $\log \frac{1}{1-\lambda_{\min}^2(\boldsymbol{K})} \to 0$ as $n \to \infty$. In contrast, as can be seen later (in Corollary 2), if $f^*$ lies in the span of a small number of eigenspaces of the integral operator in (11), then we can choose $\eta = \Theta(1)$, choose $c_0$ to be a value that is determined by the target function $f^*$ and the distribution $\rho$ only, and choose $c_1 = \Theta(\frac{1}{\sqrt{n}})$. Thus, the resulting convergence rate $\log \frac{1}{1-\eta c_0}$ does not approach 0 as $n \to \infty$. The additive term $c_1 = \Theta(1/\sqrt{n})$ arises from the fact that only finitely many data tuples are available. Both the proof of Theorem 3 and the proofs in [DZPS18, ADH+19, AZLL18] are based on the observation that when the network is sufficiently over-parameterized, the sign changes (activation pattern changes) of the hidden neurons are sparse. Different from [DZPS18, ADH+19], our proof does not use $\lambda_{\min}(\boldsymbol{K})$; see Appendix E for details.

It remains to show, with high probability, (13) in Theorem 3 holds with properly chosen $c_0$ and $c_1$. By the spectral theorem [DS63, Theorem 4, Chapter X.3] and [RBV10], $L_{\mathcal{K}}$ has a spectrum with *distinct* eigenvalues $\mu_1 > \mu_2 > \cdots$ [4] such that

$$L_{\mathcal{K}} = \sum_{i \ge 1} \mu_i P_{\mu_i}, \quad \text{with } P_{\mu_i} := \frac{1}{2\pi \mathsf{i}} \int_{\Gamma_{\mu_i}} (\gamma \mathcal{I} - L_{\mathcal{K}})^{-1} \mathrm{d}\gamma,$$

where $P_{\mu_i} : L^2(\mathcal{S}^{d-1}, \rho) \to L^2(\mathcal{S}^{d-1}, \rho)$ is the *orthogonal projection operator* onto the eigenspace associated with eigenvalue $\mu_i$; here (1) $\mathsf{i}$ is the imaginary unit, and (2) the integral can be taken over any closed simple rectifiable curve (with positive direction) $\Gamma_{\mu_i}$ containing $\mu_i$ only and no other distinct eigenvalue. In other words, $P_{\mu_i} f$ is the function obtained by projecting function $f$ onto the eigenspaces of the integral operator $L_{\mathcal{K}}$ associated with $\mu_i$.

Given an $\ell \in \mathbb{N}$, let $m_\ell$ be the sum of the multiplicities of the first $\ell$ nonzero top eigenvalues of $L_{\mathcal{K}}$. That is, $m_1$ is the multiplicity of $\mu_1$ and $(m_2 - m_1)$ is the multiplicity of $\mu_2$. By definition,

$$\lambda_{m_\ell} = \mu_\ell \ne \mu_{\ell+1} = \lambda_{m_\ell+1}, \quad \forall\, \ell.$$

**Theorem 4.** *For any $\ell \ge 1$ such that $\mu_i > 0$, for $i \le \ell$, let*

$$\epsilon(f^*, \ell) := \sup_{x \in \mathcal{S}^{d-1}} \left| f^*(x) - \left( \sum_{1 \le i \le \ell} P_{\mu_i} f^* \right)(x) \right|$$

*be the approximation error of the span of the eigenspaces associated with the first $\ell$ distinct eigenvalues. Then given $\delta \in (0, \frac{1}{4})$ and $T > 0$, if $n > \frac{256 \log \frac{2}{\delta}}{(\lambda_{m_\ell} - \lambda_{m_\ell+1})^2}$ and*

$m \geq \frac{32}{c_1^2}\left(\left(\frac{1}{c_0} + 2\eta T c_1\right)^4 + 4\log\frac{4n}{\delta}\left(\frac{1}{c_0} + 2\eta T c_1\right)^2\right)$ *with* $c_0 = \frac{3}{4}\lambda_\ell$ *and* $c_1 = \epsilon(f^*, \ell)$, *then with probability* $\geq (1 - 3\delta)$, *for all* $t \leq T$:

$$\left\|\frac{1}{\sqrt{n}}\left(\boldsymbol{y} - \widehat{\boldsymbol{y}}(t)\right)\right\| \leq \left(1 - \frac{3}{4}\eta\lambda_{m_\ell}\right)^t + \frac{16\sqrt{2}\sqrt{\log\frac{2}{\delta}}}{(\lambda_{m_\ell} - \lambda_{m_\ell+1})\sqrt{n}} + 2\sqrt{2}\epsilon(f^*, \ell).$$

Since $\lambda_{m_\ell}$ is determined by $f^*$ and $\rho$ only, with $\eta = 1$, the convergence rate $\log\frac{1}{1 - \frac{3}{4}\lambda_{m_\ell}}$ is constant w. r. t. $n$.

**Remark 1** (Early stopping). In Theorems 3 and 4, the derived lower bounds of $m$ grow in $T$. To control $m$, we need to terminate the GD training at some "reasonable" $T$. Fortunately, $T$ is typically small. To see this, note that $\eta$, $c_0$, and $c_1$ are independent of $t$. By (13) and (15) we know $\left\|\frac{1}{\sqrt{n}}\left(\boldsymbol{y} - \widehat{\boldsymbol{y}}(t)\right)\right\|$ decreases to $\Theta(c_1)$ in $(\log\frac{1}{c_1} / \log\frac{1}{1-\eta c_0})$ iterations provided that $(\log\frac{1}{c_1} / \log\frac{1}{1-\eta c_0}) \leq T$. Thus, to guarantee $\left\|\frac{1}{\sqrt{n}}\left(\boldsymbol{y} - \widehat{\boldsymbol{y}}(t)\right)\right\| = O(c_1)$, it is enough to terminate GD at iteration $T = \Theta(\log\frac{1}{c_1} / \log\frac{1}{1-\eta c_0})$. Similar to us, early stopping is adopted in [AZLL18, LSO19], and is commonly adopted in practice.

**Corollary 2** (zero–approximation error). *Suppose there exists* $\ell$ *such that* $\mu_i > 0$, *for* $i \leq \ell$, *and* $\epsilon(f^*, \ell) = 0$. *Then let* $\eta = 1$ *and* $T = \frac{\log n}{-\log(1 - \frac{3}{4}\lambda_{m_\ell})}$. *For a given* $\delta \in (0, \frac{1}{4})$, *if* $n > \frac{256\log\frac{2}{\delta}}{(\lambda_{m_\ell} - \lambda_{m_\ell+1})^2}$ *and* $m \gtrsim (n\log n)\left(\frac{1}{\lambda_{m_\ell}^4} + \frac{\log^4 n \log^2 \frac{1}{\delta}}{(\lambda_{m_\ell} - \lambda_{m_\ell+1})^2 n^2 \lambda_{m_\ell}^4}\right)$, *then with probability* $\geq (1 - 3\delta)$, *for all* $t \leq T$:

$$\left\|\frac{1}{\sqrt{n}}\left(\boldsymbol{y} - \widehat{\boldsymbol{y}}(t)\right)\right\| \leq (1 - \frac{3\lambda_{m_\ell}}{4})^t + \frac{16\sqrt{2\log 2/\delta}}{\sqrt{n}\,(\lambda_{m_\ell} - \lambda_{m_\ell+1})}.$$

Corollary 2 says that for fixed $f^*$ and fixed distribution $\rho$, nearly-linear network over-parameterization $m = \Theta(n\log n)$ is enough for GD method to converge exponentially fast as long as $\frac{1}{\delta} = O(\mathsf{poly}(n))$. Corollary 2 follow immediately from Theorem 4 by specifying the relevant parameters such as $\eta$ and $T$. To the best of our knowledge, this is the first result showing sufficiency of nearly-linear network over-parameterization. Note that $(\lambda_{m_\ell} - \lambda_{m_\ell+1}) > 0$ is the eigengap between the $\ell$–th and $(\ell+1)$–th largest distinct eigenvalues of the integral operator, and is irrelevant to $n$. Thus, for fixed $f^*$ and $\rho$, $c_1 = \Theta\left(\sqrt{\log\frac{1}{\delta}/n}\right)$.

## 4  Application to Uniform Distribution and Polynomials

We illustrate our general results by applying them to the setting where the target functions are polynomials and the feature vectors are uniformly distributed on the sphere $\mathcal{S}^{d-1}$.

Up to now, we implicitly incorporate the bias $b_j$ in $w_j$ by augmenting the original $w_j$; correspondingly, the data feature vector is also augmented. In this section, as we are dealing with distribution on the original feature vector, we explicitly separate out the bias from $w_j$. In particular, let $b_j^0 \sim \mathcal{N}(0, 1)$. For ease of exposition, with a little abuse of notation, we use $d$ to denote the dimension of the $w_j$ and $x$ before the above mentioned augmentation. With bias, (1) can be rewritten as $f_{\boldsymbol{W}, \boldsymbol{b}}(x) = \frac{1}{\sqrt{m}}\sum_{j=1}^m a_j \left[\langle x, w_j\rangle + b_j\right]_+$, where $\boldsymbol{b} = (b_1, \cdots, b_m)$ are the bias of the hidden neurons, and the kernel function in (12) becomes

$$\mathcal{K}(x, s) = \frac{\langle x, s\rangle + 1}{2\pi}\left(\pi - \arccos\left(\frac{1}{2}\left(\langle x, s\rangle + 1\right)\right)\right) \quad \forall\, x, s \in \mathcal{S}^{d-1}. \tag{16}$$

From Theorem 4 we know the convergence rate is determined by the eigendecomposition of the target function $f^*$ w. r. t. the eigenspaces of $L_\mathcal{K}$. When $\rho$ is the uniform distribution on $\mathcal{S}^{d-1}$, the eigenspaces of $L_\mathcal{K}$ are the spaces of homogeneous harmonic polynomials, denoted by $\mathcal{H}^\ell$ for $\ell \geq 0$. Specifically, $L_\mathcal{K} = \sum_{\ell \geq 0}\beta_\ell P_\ell$, where $P_\ell$ (for $\ell \geq 0$) is the orthogonal projector onto $\mathcal{H}^\ell$ and $\beta_\ell = \frac{\alpha_\ell \frac{d-2}{2}}{\ell + \frac{d-2}{2}} > 0$ is the associated eigenvalue $-\alpha_\ell$ is the coefficient of $\mathcal{K}(x, s)$ in the expansion into

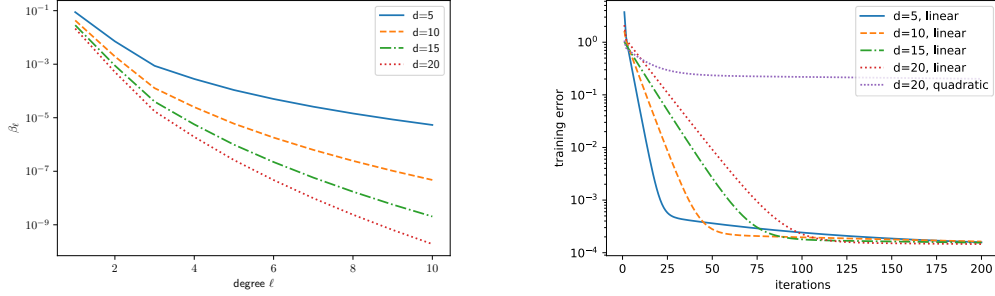

(a) Plot of $\beta_\ell$ with $\ell$ under different $d$. Here, the $\beta_\ell$ is monotonically decreasing in $\ell$.

(b) Training with $f^*$ being randomly generated linear or quadratic functions with $n = 1000$, $m = 2000$.

Figure 2: Application to uniform distribution and polynomials.

Gegenbauer polynomials. Note that $\mathcal{H}^\ell$ and $\mathcal{H}^{\ell'}$ are orthogonal when $\ell \neq \ell'$. See appendix G for relevant backgrounds on harmonic analysis on spheres.

Explicit expression of eigenvalues $\beta_\ell > 0$ is available; see Fig. 2a for an illustration of $\beta_\ell$. In fact, there is a line of work on efficient computation of the coefficients of Gegenbauer polynomials expansion [CI12].

If the target function $f^*$ is a standard polynomial of degree $\ell^*$, by [Wan, Theorem 7.4], we know $f^*$ can be perfectly projected onto the direct sum of the spaces of homogeneous harmonic polynomials up to degree $\ell^*$. The following corollary follows immediately from Corollary 2.

**Corollary 3.** *Suppose $f^*$ is a degree $\ell^*$ polynomial, and the feature vector $x_i$'s are i.i.d. generated from the uniform distribution over $\mathcal{S}^{d-1}$. Let $\eta = 1$, and $T = \Theta(\log n)$. For a given $\delta \in (0, \frac{1}{4})$, if $n = \Theta\left(\log \frac{1}{\delta}\right)$ and $m = \Theta(n \log n \log^2 \frac{1}{\delta})$, then with probability at least $1 - \delta$, for all $t \leq T$:*

$$\left\| \frac{1}{\sqrt{n}} \left( \boldsymbol{y} - \widehat{\boldsymbol{y}}(t) \right) \right\| \leq \left( 1 - \frac{3c_0}{4} \right)^t + \Theta(\sqrt{\frac{\log 1/\delta}{n}}), \quad \text{where } c_0 = \min \left\{ \beta_{\ell^*}, \beta_{\ell^*+1} \right\}.$$

For ease of exposition, in the above corollary, $\Theta(\cdot)$ hides dependence on quantities such as eigengaps – as they do not depend on $n$, $m$, and $\delta$. Corollary 3 and $\beta_\ell$ in Fig. 2a together suggest that the convergence rate decays with both the dimension $d$ and the polynomial degree $\ell$. This is validated in Fig. 2a. It might be unfair to compare the absolute values of training errors since $f^*$ are different. Nevertheless, the convergence rates can be read from slope in logarithmic scale. We see that the convergence slows down as $d$ increases, and learning a quadratic function is slower than learning a linear function.

Next we present the explicit expression of $\beta_\ell$. For ease of exposition, let $h(u) := \mathcal{K}(x, s)$ where $u = \langle x, s \rangle$. By [CI12, Eq. (2.1) and Theorem 2], we know

$$\beta_\ell = \frac{d-2}{2} \sum_{k=0}^{\infty} \frac{h_{\ell+2k}}{2^{\ell+2k} k! \left( \frac{d-2}{2} \right)_{\ell+k+1}}, \tag{17}$$

where $h_\ell := h^{(\ell)}(0)$ is the $\ell$-th order derivative of $h$ at zero, and the *Pochhammer symbol* $(a)_k$ is defined recursively as $(a)_0 = 1$, $(a)_k = (a + k - 1)(a)_{k-1}$ for $k \in \mathbb{N}$. By a simple induction, it can be shown that $h_0 = h^{(0)}(0) = 1/3$, and for $k \geq 1$,

$$h_k = \frac{1}{2} \mathbf{1}_{\{k=1\}} - \frac{1}{\pi 2^k} \left( k \left( \arccos 0.5 \right)^{(k-1)} + 0.5 \left( \arccos 0.5 \right)^{(k)} \right), \tag{18}$$

where the computation of the higher-order derivative of $\arccos$ is standard. It follows from (17) and (18) that $\beta_\ell > 0$, and $\beta_{2\ell} > \beta_{2(\ell+1)}$ and $\beta_{2\ell+1} > \beta_{2\ell+3}$ for all $\ell \geq 0$. However, an analytic order among $\beta_\ell$ is unclear, and we would like to explore this in the future.

## Footnotes

[1]Our analysis might be adapted to other initialization schemes, such as He initialization, with $m = \Omega(n^2)$. Nevertheless, the more stringent requirement on $m$ might only be an artifact of our analysis.

[2]The simplification assumption that the second layer is fixed is also adopted in [DZPS18, ADH+19]. Similar frozen assumption is adopted in [ZCZG18, AZLS18]. We do agree this assumption might restrict the applicability of our results. Nevertheless, even this setting is not well-understood despite the recent intensive efforts.

[3] Though a refined analysis of that in [DZPS18] is given by [ADH$^+$19, Theorem 4.1], the analysis crucially relies on the convergence rate in (8).

[4] The sequence of distinct eigenvalues can possibly be of finite length. In addition, the sequences of $\mu_i$'s and $\lambda_i$'s (in Theorem 2) are different, the latter of which consists of repetitions.

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
