[Supplementary Material]

# Supplementary Materials

## A Related Work

The volume of the literature on neural networks is growing rapidly, and we cannot hope to do justice to this large body of related work. Here we sample an incomplete list of works that are most relevant to this work.

There have been intensive efforts in proving the (global) convergence of the simple first-order methods such as (stochastic) gradient descent [BG17, LY17, ZSJ$^+$17], where the true function that generates the responses/labels is a two-layer neural network of the same size as the neural network candidates. Notably, in this line of work, it is typically assumed that $m \leq d$.

Over-parameterized neural networks are observed to enjoy smaller training errors and even smaller generalization errors [ZBH$^+$16, LL18]. Allen-Zhu et al. [AZLL18] considered the setting where the true network is much smaller than the candidate networks, and showed that searching among over-parametrized network candidates smoothes the optimization trajectory and enjoys a strongly-convex-like behavior. Similar results were shown in [DZPS18, ADH$^+$19, OS19]. In particular, it was shown in an inspiring work [DZPS18] that when $m = \Omega(n^6)$ and the minimum eigenvalue of some Gram matrix is positive, then randomly initialized GD can find an optimal neural network, under squared loss, at a linear rate. However, the involved minimum eigenvalue scales in $n$, and the impacts of this scaling on the convergence and the corresponding convergence rate were overlooked in [DZPS18]. Unfortunately, taking such scaling into account, their convergence rate approaches 0 as $n \to \infty$; we formally show this in Theorem 2 and Corollary 1. Recently, [OS19] showed that when $m = \Omega(n^2)$, the empirical risk (training error) goes to zero at a linear rate $(1 - c\frac{d}{n})^t = \exp\left(-t\log(1/(1 - c\frac{d}{n}))\right)$, where $t$ is the GD iteration and $c > 0$ is some absolute constant; see [OS19, Corollaries 2.2 and 2.4] for details. Here $\log(1/(1 - c\frac{d}{n}))$ is the convergence rate. It is easy to see that $\log(1/(1 - c\frac{d}{n})) \to 0$ as $n$ increases.

For deep networks (which contain more than one hidden layer), the (global) convergence of (S)GD are shown in [ZCZG18, DLL$^+$18, AZLS18] with different focuses and characterizations of over-parameterization sufficiency. In particular, [ZCZG18] studied the binary classification problem and showed that (S)GD can find a global minimum provided that the feature vectors with different labels are well separated and $m = \Omega(\text{poly}(n, L))$, where $L$ is the number of hidden layers. Allen-Zhu et al. [AZLS18] considered the regression problem and showed similar over-parameterization sufficiency. The over-parameterization sufficiency in terms of its scaling in $n$ is significantly improved in [DLL$^+$18] without considering the scaling of the minimum eigenvalue of the Gram matrix in $n$.

All the above recent progress is established on the common observation that when the network is sufficiently over-parameterized, during training, the network weights are mainly stay within a small perturbation region centering around the initial weights. In fact, the over-parameterization sufficiency that ensures the above mild weight changes is often referred to as NTK region; see [JGH18] for details. Recently, a handful of work studied linearized neural networks in high dimension [GMMM19, YS19, VW18]. Since we consider fixed $d$, our results are not directly comparable to that line of work.

## B Existing Generalization Bound

Though a generalization bound is given in [ADH$^+$19, Theorem 5.1 and Corollary 5.2], it is unclear how their bound scales in $n$. In particular, the dominating term of their bound is $\sqrt{\frac{2\boldsymbol{y}^\top (n\boldsymbol{H})^{-1}\boldsymbol{y}}{n}}$. Here, the matrix $\boldsymbol{H}$ is defined w.r.t. the training dataset $\{(x_i, y_i) : 1 \leq i \leq n\}$ and the $\ell_2$ norm of the response vector $\boldsymbol{y}$ grows with $n$. As a result of this, the scaling of the magnitude of $\boldsymbol{y}^\top (n\boldsymbol{H})^{-1} \boldsymbol{y}$ in $n$ is unclear. Recall that $y_i = \Theta(1)$ for $i = 1, \cdots, n$; thus, $\|\boldsymbol{y}\| = \Theta(\sqrt{n})$. If we do not care about the structure of the target function $f^*$ and allow $\frac{\boldsymbol{y}}{\sqrt{n}}$ to be the eigenvector associated with the least eigenvalue of $\boldsymbol{H}$, then $\sqrt{\frac{2\boldsymbol{y}^\top (n\boldsymbol{H})^{-1}\boldsymbol{y}}{n}}$ might not decrease to zero as $n \to \infty$. This is because

$$\sqrt{\frac{2\boldsymbol{y}^\top (n\boldsymbol{H})^{-1}\boldsymbol{y}}{n}} = \sqrt{2\left(\frac{\boldsymbol{y}}{\sqrt{n}}\right)^\top (n\boldsymbol{H})^{-1}\left(\frac{\boldsymbol{y}}{\sqrt{n}}\right)} = \Theta\left((\lambda_{\min}(n\boldsymbol{H}))^{-\frac{1}{2}}\right).$$

As illustrated by Fig. 3, even when $\rho$ is the uniform distribution, $(\lambda_{\min}(n\boldsymbol{H}))^{-\frac{1}{2}}$ does not approach zero as $n$ increases. In general, without specifying the structure of the target function $f^*$, in the presence of the randomness of data generation and the network initialization, it is unclear which eigenvalues of $\boldsymbol{H}$ determines the generalization capability of the learned neural network.

Figure 3: Plot of $(\lambda_{\min}(n\boldsymbol{H}))^{-\frac{1}{2}}$ under different sample sizes. Here the feature vectors are generated from the uniform distribution on the unit sphere.

## C  Proof of Theorem 1

We use the following proposition in proving Theorem 1.

**Proposition 1.** *It is true that for any $j \in [m]$, $i \in [n]$, and $t \geq 0$,*

$$\frac{\eta a_j}{n\sqrt{m}} \sum_{i'=1}^{n} (y_{i'} - \widehat{y}_{i'}(t)) \langle x_i, x_{i'} \rangle \mathbf{1}_{\{\langle w_j^t, x_{i'} \rangle > 0\}} \mathbf{1}_{\{\langle w_j^t, x_i \rangle > 0\}}$$

$$\leq \left[ \langle w_j^{t+1}, x_i \rangle \right]_+ - \left[ \langle w_j^t, x_i \rangle \right]_+ \tag{19}$$

$$\leq \frac{\eta a_j}{n\sqrt{m}} \sum_{i'=1}^{n} (y_{i'} - \widehat{y}_{i'}(t)) \langle x_i, x_{i'} \rangle \mathbf{1}_{\{\langle w_j^t, x_{i'} \rangle > 0\}} \mathbf{1}_{\{\langle w_j^{t+1}, x_i \rangle > 0\}}. \tag{20}$$

*Proof.* From (5), we have

$$\langle w_j^{t+1}, x_i \rangle - \langle w_j^t, x_i \rangle = \frac{\eta a_j}{n\sqrt{m}} \sum_{i'=1}^{n} (y_{i'} - \widehat{y}_{i'}(t)) \langle x_i, x_{i'} \rangle \mathbf{1}_{\{\langle w_j^t, x_{i'} \rangle > 0\}}. \tag{21}$$

Then the conclusion follows from the fact that

$$\mathbf{1}_{\{a>0\}}(b - a) \leq \ [b]_+ - [a]_+ \ \leq \mathbf{1}_{\{b>0\}}(b - a), \ \ \forall \, a, b. \tag{22}$$

$\square$

**Remark 2.** The inequality in (22) can be extended to a general family of activation function $\sigma$ if

$$\sigma'(a)(b - a) \leq \sigma(b) - \sigma(a) \leq \sigma'(b)(b - a), \quad \forall \, a, b,$$

where $\sigma'(\cdot)$ is the derivative of $\sigma$. For ReLu activation, the right derivative is used.

**Proof of Theorem 1.** Recall from (4) that for $t \geq 0$,

$$\widehat{y}_i(t + 1) = \frac{1}{\sqrt{m}} \sum_{j=1}^{m} a_j \left[ \langle w_j^{t+1}, x \rangle \right]_+ = \frac{1}{\sqrt{m}} \sum_{j \in \mathcal{A}} \left[ \langle w_j^{t+1}, x \rangle \right]_+ - \frac{1}{\sqrt{m}} \sum_{j \in \mathcal{B}} \left[ \langle w_j^{t+1}, x \rangle \right]_+.$$

Thus,

$$\widehat{y}_i(t+1) - \widehat{y}_i(t) = \frac{1}{\sqrt{m}} \sum_{j\in\mathcal{A}} \left( [\langle w_j^{t+1}, x\rangle]_+ - [\langle w_j^t, x\rangle]_+ \right) - \frac{1}{\sqrt{m}} \sum_{j\in\mathcal{B}} \left( [\langle w_j^{t+1}, x\rangle]_+ - [\langle w_j^t, x\rangle]_+ \right)$$

$$\overset{(a)}{\leq} \frac{\eta}{nm} \sum_{j\in\mathcal{A}} \sum_{i'=1}^n (y_{i'} - \widehat{y}_{i'}(t)) \langle x_i, x_{i'}\rangle \mathbf{1}_{\{\langle w_j^t, x_{i'}\rangle > 0\}} \mathbf{1}_{\{\langle w_j^{t+1}, x_i\rangle > 0\}}$$

$$+ \frac{\eta}{nm} \sum_{j\in\mathcal{B}} \sum_{i'=1}^n (y_{i'} - \widehat{y}_{i'}(t)) \langle x_i, x_{i'}\rangle \mathbf{1}_{\{\langle w_j^t, x_{i'}\rangle > 0\}} \mathbf{1}_{\{\langle w_j^t, x_i\rangle > 0\}}$$

$$= \eta \sum_{i'=1}^n \left( \widetilde{\boldsymbol{H}}_{ii'}^+(t+1) + \boldsymbol{H}_{ii'}^-(t+1) \right) (y_{i'} - \widehat{y}_{i'}(t)),$$

where inequality (a) follows from Proposition 1. Thus,

$$y_i - \widehat{y}_i(t+1) \geq y_i - \widehat{y}_i(t) - \eta \sum_{i'=1}^n \left( \widetilde{\boldsymbol{H}}_{ii'}^+(t+1) + \boldsymbol{H}_{ii'}^-(t+1) \right) (y_{i'} - \widehat{y}_{i'}(t)),$$

whose matrix form is $(\boldsymbol{y} - \widehat{\boldsymbol{y}}(t+1)) \geq \left( \boldsymbol{I} - \eta \left( \widetilde{\boldsymbol{H}}^+(t+1) + \boldsymbol{H}^-(t+1) \right) \right) (\boldsymbol{y} - \widehat{\boldsymbol{y}}(t))$, proving the lower bound in Theorem 1. The upper bound in Theorem 1 can be obtained analogously.

$\square$

## D   Proof of Theorem 2

Let $\lambda_1 \geq \lambda_2 \geq \dots$ be the spectrum of $L_\mathcal{K}$ defined in (11), whose existence is given by the spectral theorem [DS63, Theorem 4, Chapter X.3]. Recall that

$$\boldsymbol{H}_{ii'} = \frac{1}{nm} \langle x_i, x_{i'}\rangle \sum_{j=1}^m \mathbf{1}_{\{\langle w_j^0, x_{i'}\rangle > 0\}} \mathbf{1}_{\{\langle w_j^0, x_i\rangle > 0\}}$$

is a random $n \times n$ matrix, where the randomness comes from (1) the data randomness $\{(x_i, y_i) : 1 \leq i \leq n\}$ and (2) the network initialization randomness. Thus, $\widetilde{\lambda}_i$ for $1 \leq i \leq n$ are random. Notably, $\boldsymbol{K} = \mathbb{E}[\boldsymbol{H}]$ is still random as the data randomness remains. Denote the spectrum of $\boldsymbol{K}$ as $\widehat{\lambda}_1 \geq \dots \geq \widehat{\lambda}_n$. By [RBV10, Proposition 10], with probability at least $1 - \frac{\delta}{2}$ over data generation,

$$\sup_i |\lambda_i - \widehat{\lambda}_i| \leq \sqrt{\frac{8\log(4/\delta)}{n}}. \tag{23}$$

For a given dataset $x_1, \dots, x_n \in \mathcal{S}^{d-1}$, by Hoeffding's inequality and the union bound, with probability at least $1 - \frac{\delta}{2}$ over network initialization,

$$\|\boldsymbol{H} - \boldsymbol{K}\|_F = \|\boldsymbol{H} - \mathbb{E}\boldsymbol{H}\|_F \leq \sqrt{\frac{\log(4n^2/\delta)}{m}}. \tag{24}$$

Then, it follows from Weyl's inequality that

$$\sup_i |\widetilde{\lambda}_i - \widehat{\lambda}_i| \leq \sqrt{\frac{\log(4n^2/\delta)}{m}}. \tag{25}$$

We conclude (9) by combining (23) and (25). Letting $\delta = \frac{1}{n}$, we have, with probability $1 - \frac{1}{n}$,

$$0 \leq \lambda_{\min}(\boldsymbol{H}) \leq \lambda_n + \sqrt{\frac{\log(4n^3)}{m}} + \sqrt{\frac{8\log(4n)}{n}},$$

where the right-hand side vanishes with $n$. Thus, let $n \to \infty$, we have $1 - \frac{1}{n} \to 1$, and

$$\lim_{n\to\infty} \lambda_n + \lim_{n\to\infty} \sqrt{\frac{\log(4n^3)}{m}} + \lim_{n\to\infty} \sqrt{\frac{8\log(4n)}{n}} = 0,$$

proving the theorem.

# E  Proof of Theorem 3

We prove Theorem 3 via two steps: (1) We first bound the perturbation terms. (2) Then, we prove Theorem 3 via an induction argument.

## E.1  Bounding the perturbation

For ease of exposition, let

$$\boldsymbol{H}(t) := \left(\boldsymbol{H}^+(t) + \boldsymbol{H}^-(t)\right), \tag{26}$$

$$\boldsymbol{M}(t) := \left(\widetilde{\boldsymbol{H}}^-(t) - \boldsymbol{H}^-(t)\right), \tag{27}$$

$$\boldsymbol{L}(t) := \left(\widetilde{\boldsymbol{H}}^+(t) - \boldsymbol{H}^+(t)\right). \tag{28}$$

**Lemma 1.** *Choose* $0 < \eta < 1$. *Then for any* $t \geq 0$, *it holds that*

$$\|\boldsymbol{y} - \widehat{\boldsymbol{y}}(t+1)\| \leq \left\|(\boldsymbol{I} - \eta\boldsymbol{K})^{t+1}\,\boldsymbol{y}\right\| + \eta \sum_{r=2}^{t+2} \|(\boldsymbol{K} - \boldsymbol{H}(r-1))\| \left\|(\boldsymbol{I} - \eta\boldsymbol{K})^{r-2}\,\boldsymbol{y}\right\|$$

$$+ \eta \sum_{r=2}^{t+2} \left(\|\boldsymbol{M}(r-1)\| + \|\boldsymbol{L}(r-1)\|\right) \|(\boldsymbol{y} - \widehat{\boldsymbol{y}}(r-2))\|.$$

*Proof.* Let $\boldsymbol{\epsilon}(t+1) := (\boldsymbol{y} - \widehat{\boldsymbol{y}}(t+1)) - (\boldsymbol{I} - \eta\boldsymbol{H}(t+1))(\boldsymbol{y} - \widehat{\boldsymbol{y}}(t))$, for $t \geq 0$, i.e.,

$$\boldsymbol{y} - \widehat{\boldsymbol{y}}(t+1) = (\boldsymbol{I} - \eta\boldsymbol{H}(t+1))(\boldsymbol{y} - \widehat{\boldsymbol{y}}(t)) + \boldsymbol{\epsilon}(t+1), \quad \forall\, t \geq 0. \tag{29}$$

It follows from Theorem 1 that

$$\|\boldsymbol{\epsilon}(t+1)\| \leq \eta(\|\boldsymbol{M}(t+1)\| + \|\boldsymbol{L}(t+1)\|) \|\boldsymbol{y} - \widehat{\boldsymbol{y}}(t)\|. \tag{30}$$

Expanding (29) over $t$, we have

$$\boldsymbol{y} - \widehat{\boldsymbol{y}}(t+1) = \left[\prod_{r=1}^{t+1} (\boldsymbol{I} - \eta\boldsymbol{H}(r))\right] (\boldsymbol{y} - \widehat{\boldsymbol{y}}(0)) + \sum_{r=2}^{t+2} \left[\prod_{k=r}^{t+1} (\boldsymbol{I} - \eta\boldsymbol{H}(k))\right] \boldsymbol{\epsilon}(r-1), \tag{31}$$

where $\prod_{r=k}^{t+1} (\boldsymbol{I} - \eta\boldsymbol{H}(r)) := (\boldsymbol{I} - \eta\boldsymbol{H}(t+1)) \times \cdots \times (\boldsymbol{I} - \eta\boldsymbol{H}(k))$ for $k \leq t+1$ is a backward matrix product, and $\prod_{k=t+2}^{t+1} (\boldsymbol{I} - \eta\boldsymbol{H}(k)) := \boldsymbol{I}$. Recall that $\widehat{\boldsymbol{y}}(0) = \boldsymbol{0}$. Eq. (31) can be simplified as

$$\boldsymbol{y} - \widehat{\boldsymbol{y}}(t+1) = \left[\prod_{r=1}^{t+1} (\boldsymbol{I} - \eta\boldsymbol{H}(r))\right] \boldsymbol{y} + \sum_{r=2}^{t+2} \left[\prod_{k=r}^{t+1} (\boldsymbol{I} - \eta\boldsymbol{H}(k))\right] \boldsymbol{\epsilon}(r-1).$$

In addition, it can be shown by a simple induction that

$$\left[\prod_{r=1}^{t+1} (\boldsymbol{I} - \eta\boldsymbol{H}(r))\right] = (\boldsymbol{I} - \eta\boldsymbol{K})^{t+1} + \eta \sum_{r=2}^{t+2} \left[\prod_{k=r}^{t+1} (\boldsymbol{I} - \eta\boldsymbol{H}(k))\right] (\boldsymbol{K} - \boldsymbol{H}(r-1))(\boldsymbol{I} - \eta\boldsymbol{K})^{r-2}.$$

Thus, we have

$$\boldsymbol{y} - \widehat{\boldsymbol{y}}(t+1) = (\boldsymbol{I} - \eta\boldsymbol{K})^{t+1}\,\boldsymbol{y} + \eta \sum_{r=2}^{t+2} \left[\prod_{k=r}^{t+1} (\boldsymbol{I} - \eta\boldsymbol{H}(k))\right] (\boldsymbol{K} - \boldsymbol{H}(r-1))(\boldsymbol{I} - \eta\boldsymbol{K})^{r-2}\,\boldsymbol{y}$$

$$+ \sum_{r=2}^{t+2} \left[\prod_{k=r}^{t+1} (\boldsymbol{I} - \eta\boldsymbol{H}(k))\right] \boldsymbol{\epsilon}(r-1).$$

Notably, $\|\boldsymbol{H}(k)\|^2 \leq \|\boldsymbol{H}(k)\|_{\mathrm{F}}^2 \leq 1$ for each $k \geq 1$. Choosing $0 < \eta < 1$, we have $\|\boldsymbol{I} - \eta\boldsymbol{H}(k)\| \leq 1$ for $k \geq 1$. With this fact and (30), we conclude Lemma 1.

$\square$

For each $i \in [n]$ and $t \geq 0$, let

$$\mathcal{F}(x_i, t) := \left\{ j : \exists\, 0 \leq k \leq t \text{ s.t. } \mathbf{1}_{\left\{\langle w_j^k, x_i \rangle > 0\right\}} \neq \mathbf{1}_{\left\{\langle w_j^0, x_i \rangle > 0\right\}} \right\}. \tag{32}$$

be the set of hidden neurons that have ever flipped their signs by iteration $t$.

**Lemma 2.** *Choose $0 < \eta < 1$. The following holds for all $t \geq 0$:*

$$\max \left\{ \|\boldsymbol{M}(t)\| + \|\boldsymbol{L}(t)\|, \ \|\boldsymbol{H} - \boldsymbol{H}(t)\| \right\} \leq \sqrt{\frac{4}{m^2 n} \sum_{i=1}^{n} |\mathcal{F}(x_i, t)|^2}.$$

*Proof.* We bound $\boldsymbol{M}(t)$ as

$$\|\boldsymbol{M}(t)\|^2 \leq \|\boldsymbol{M}(t)\|_{\mathrm{F}}^2 = \sum_{i=1}^{n} \sum_{i'=1}^{n} \boldsymbol{M}_{ii'}^2(t)$$

$$\leq \frac{1}{m^2 n^2} \sum_{i=1}^{n} \sum_{i'=1}^{n} \left( \langle x_i, x_{i'} \rangle \sum_{j \in \mathcal{F}(x_i,t) \cap \mathcal{A}} \mathbf{1}_{\left\{\langle w_j^t, x_{i'} \rangle + b_j^t\right\}} \right)^2$$

$$\leq \frac{1}{m^2 n} \sum_{i=1}^{n} |\mathcal{F}(x_i, t)|^2. \tag{33}$$

Similarly, $\|\boldsymbol{L}(t)\|^2 \leq \frac{1}{m^2 n} \sum_{i=1}^{n} |\mathcal{F}(x_i, t)|^2$ and $\|\boldsymbol{H} - \boldsymbol{H}(t)\|^2 \leq \frac{4}{m^2 n} \sum_{i=1}^{n} |\mathcal{F}(x_i, t)|^2$. $\square$

**Lemma 3.** *Fix a dataset $\{(x_i, y_i) : 1 \leq i \leq n\}$. For any $R > 0$ and $\delta \in (0, \frac{1}{4})$, with probability at least $1 - \delta$ over network initialization,*

$$\|\boldsymbol{K} - \boldsymbol{H}\| + \sqrt{\frac{4}{m^2 n} \sum_{i=1}^{n} \left( \sum_{j=1}^{m} \mathbf{1}_{\left\{|\langle w_j^0, x_i \rangle| \leq R\right\}} \right)^2} \leq \frac{4R}{\sqrt{2\pi}} + 4\sqrt{\frac{\log(4n/\delta)}{m}}. \tag{34}$$

*Proof.* Since $w_j^0 \sim \mathcal{N}(\mathbf{0}, \boldsymbol{I})$ and $x_i \in \mathcal{S}^{d-1}$, it is true that $\langle w_j^0, x_i \rangle \sim \mathcal{N}(0, 1)$. Thus, $\mathbb{E}\left[\mathbf{1}_{\left\{|\langle w_j^0, x_i \rangle| \leq R\right\}}\right] = \mathbb{P}\left\{|\langle w_j^0, x_i \rangle| \leq R\right\} < \frac{2R}{\sqrt{2\pi}}$ holds for any $R > 0$. By Hoeffding's inequality and union bound, we have, with probability at least $1 - \frac{\delta}{2}$,

$$\frac{1}{m^2 n} \sum_{i=1}^{n} \left( \sum_{j=1}^{m} \mathbf{1}_{\left\{|\langle w_j^0, x_i \rangle| \leq R\right\}} \right)^2 \leq \left( \frac{2R}{\sqrt{2\pi}} + \sqrt{\frac{\log(4n/\delta)}{m}} \right)^2. \tag{35}$$

In addition, we have shown in (24) that with probability at least $1 - \frac{\delta}{2}$,

$$\|\boldsymbol{H} - \boldsymbol{K}\| \leq \sqrt{\frac{\log(4n^2/\delta)}{m}}.$$

From (35) and (24), we conclude Lemma 3. $\square$

### E.2 Finishing the proof of Theorem 3

For any $\delta \in (0, \frac{1}{4})$, let $\mathcal{E}$ be the event on which (34) holds for $R = \frac{1}{\sqrt{m}}\left(\frac{1}{c_0} + 2\eta T c_1\right)$. By Lemma 3, we know $\mathbb{P}\{\mathcal{E}\} \geq 1 - \delta$.

Conditioning on event $\mathcal{E}$ occurs, we finish proving Theorem 3 via induction. Since we assume $\mathcal{E}$ has occurred, all the relevant quantities below are deterministic. The base case $t = 0$ trivially holds. Suppose (15) is true up to $t \leq T - 1$, and it suffices to prove it for $t + 1$. By (13), we have

$$\eta \sum_{r=0}^{t} \left\| \frac{1}{\sqrt{n}} \left( \boldsymbol{I} - \eta\boldsymbol{K} \right)^r \boldsymbol{y} \right\| \leq \eta \sum_{r=0}^{t} \left( (1 - \eta c_0)^r + c_1 \right) \leq \frac{1}{c_0} + \eta T c_1. \tag{36}$$

By the induction hypothesis, we have

$$\eta \sum_{r=0}^{t} \left\| \frac{1}{\sqrt{n}} \left( \boldsymbol{y} - \widehat{\boldsymbol{y}}(r) \right) \right\| \leq \eta \sum_{r=0}^{t} ((1 - \eta c_0)^r + 2c_1) \leq \frac{1}{c_0} + 2\eta T c_1. \tag{37}$$

Also, we have $|\mathcal{F}(x_i, r)| \leq |\mathcal{F}(x_i, t+1)|$ for each $r \leq t+1$ by monotonicity. Then, applying the upper bounds in Lemma 2 and (36) – (37) into Lemma 1, we obtain that,

$$\left\| \frac{1}{\sqrt{n}} \left( \boldsymbol{y} - \widehat{\boldsymbol{y}}(t+1) \right) \right\| \leq \left( (1 - \eta c_0)^{t+1} + c_1 \right)$$

$$+ \left( \| \boldsymbol{K} - \boldsymbol{H} \| + \sqrt{ \frac{4}{m^2 n} \sum_{i=1}^{n} |\mathcal{F}(x_i, t+1)|^2 } \right) \left( \frac{2}{c_0} + 3\eta T c_1 \right). \tag{38}$$

It remains to bound the cardinality of $\mathcal{F}(x_i, t+1)$. Note that

$$\mathcal{F}(x_i, t+1) \subseteq \left\{ j : |\langle w_j^0, x_i \rangle| \leq \max_{1 \leq k \leq t+1} \| w_j^k - w_j^0 \| \right\}.$$

Since $\| x_i \| = 1$, it follows from (5) that

$$\left\| w_j^k - w_j^{k-1} \right\| \leq \frac{\eta}{n\sqrt{m}} \sum_{i=1}^{n} |y_i - \widehat{y}_i(k-1)| \leq \frac{\eta}{\sqrt{nm}} \| \boldsymbol{y} - \widehat{\boldsymbol{y}}(k-1) \|.$$

Then, by (37), we have

$$\max_{1 \leq k \leq t+1} \left\| w_j^k - w_j^0 \right\| \leq \sum_{k=1}^{t+1} \left\| w_j^k - w_j^{k-1} \right\| \leq \frac{1}{\sqrt{m}} \left( \frac{1}{c_0} + 2\eta T c_1 \right).$$

Thus, by Lemma 3, it holds that

$$\| \boldsymbol{K} - \boldsymbol{H} \| + \sqrt{ \frac{4}{m^2 n} \sum_{i=1}^{n} \left( \sum_{j=1}^{m} \mathbf{1}_{ \{ |\langle w_j^0, x_i \rangle| \leq R \} } \right)^2 } \leq \frac{4}{\sqrt{2\pi}\sqrt{m}} \left( \frac{1}{c_0} + 2\eta T c_1 \right) + 4\sqrt{ \frac{\log(4n/\delta)}{m} }. \tag{39}$$

Substituting (39) into (38), we finish the induction.

## F  Proof of Theorem 4

We prove (13) through exploring the structure of $f^*$ and using the concentration of spectral projection. In a sense, $\frac{1}{\sqrt{n}} (\boldsymbol{I} - \eta \boldsymbol{K})^t \boldsymbol{y}$ approximates $(\mathcal{I} - \eta L_{\mathcal{K}})^t f^*$ w. r. t. some properly chosen norm. Here $\mathcal{I}$ is the identity operator, i.e., $\mathcal{I}f = f$ for each $f \in L^2(\mathcal{S}^{d-1}, \rho)$. Theorem 4 follows immediately from Theorem 3 and the following lemma.

**Lemma 4.** *For any $\ell \geq 1$ such that $\mu_i > 0$, for $i = 1, \cdots, \ell$, let*

$$\epsilon(f^*, \ell) := \sup_{x \in \mathcal{S}^{d-1}} \left| f^*(x) - \left( \sum_{1 \leq i \leq \ell} P_{\mu_i} f^* \right)(x) \right|.$$

*Then given $\delta \in (0, \frac{1}{4})$, if $n > \frac{256 \log \frac{2}{\delta}}{(\lambda_{m_\ell} - \lambda_{m_\ell + 1})^2}$, with probability at least $1 - 2\delta$ it holds that*

$$\left\| \frac{1}{\sqrt{n}} (\boldsymbol{I} - \eta \boldsymbol{K})^t \boldsymbol{y} \right\| \leq \left( 1 - \frac{3}{4} \eta \lambda_{m_\ell} \right)^t + \frac{8\sqrt{2}\sqrt{\log \frac{2}{\delta}}}{(\lambda_{m_\ell} - \lambda_{m_\ell + 1})\sqrt{n}} + \sqrt{2} \epsilon(\ell, f^*).$$

*Proof.* Since $\boldsymbol{K}$ is symmetric, we have $\boldsymbol{K} = \sum_{i=1}^{n} \widehat{\lambda}_i \widehat{u}_i \widehat{u}_i^\top$, where $\widehat{\lambda}_i$s are in an non-increasing order, $0 \leq \widehat{\lambda}_i \leq 1$, and $\| \widehat{u}_i \| = 1$. For each $i$, define a function $\widehat{\phi}_i$ over $\{ x_k : k \in [n] \}$ by

$\widehat{\phi}_i(x_k) = \sqrt{n}\widehat{u}_i(k)$ for $k \in [n]$. Let $\rho(n)$ be the empirical distribution of $\{x_k : k \in [n]\}$. Define $\langle \cdot, \cdot \rangle_{\rho(n)}$ as

$$\langle f, g \rangle_{\rho(n)} := \frac{1}{n} \sum_{k=1}^{n} f(x_k) g(x_k). \tag{40}$$

Notably, $\langle \cdot, \cdot \rangle_{\rho(n)}$ is similar to that of $\langle \cdot, \cdot \rangle_\rho$ but with a different measure. By definition $\left\{ \widehat{\phi}_i : 1 \leq i \leq n \right\}$ is a set of $n$ orthonormal functions w.r.t. the inner product $\langle \cdot, \cdot \rangle_{\rho(n)}$. It holds that

$$\frac{1}{\sqrt{n}} \left( \boldsymbol{I} - \eta \boldsymbol{K} \right)^t \boldsymbol{y} = \frac{1}{\sqrt{n}} \sum_{i=1}^{n} (1 - \eta \widehat{\lambda}_i)^t (\widehat{u}_i^\top \boldsymbol{y}) \widehat{u}_i = \sum_{i=1}^{n} (1 - \eta \widehat{\lambda}_i)^t \langle \widehat{\phi}_i, f^* \rangle_{\rho(n)} \widehat{u}_i.$$

Henceforth, we assume that $m_\ell < n$; the case $m_\ell \geq n$ can be shown similarly with the fact that $\lambda_{m_\ell} \leq \lambda_n$. Since $\langle \widehat{\phi}_i, f^* \rangle_{\rho(n)} = \frac{1}{n} \sum_{i=1}^{n} (\widehat{u}_i^\top \boldsymbol{y})^2 = \frac{1}{n} \|\boldsymbol{y}\|^2 \leq 1$, we have

$$\left\| \frac{1}{\sqrt{n}} (\boldsymbol{I} - \eta \boldsymbol{K})^t \boldsymbol{y} \right\|^2 \leq (1 - \eta \widehat{\lambda}_{m_\ell})^{2t} + \sum_{i=m_\ell+1}^{n} \langle \widehat{\phi}_i, f^* \rangle_{\rho(n)}^2. \tag{41}$$

Next we analyze the second term in (41). Let $\phi_1, \phi_2, \ldots$ be orthonormal eigenfunctions of $L_\mathcal{K}$ with strictly positive eigenvalues $\lambda_1, \lambda_2, \cdots$, respectively. Let $\gamma_j := \langle f^*, \phi_j \rangle_\rho$. It holds that

$$\sum_{i=m_\ell+1}^{n} \left( 1 - \eta \widehat{\lambda}_i \right)^{2t} \left( \left\langle \widehat{\phi}_i, f^* \right\rangle_{\rho(n)} \right)^2 \overset{(a)}{\leq} \sum_{i=m_\ell+1}^{n} \left( \left\langle \widehat{\phi}_i, f^* \right\rangle_{\rho(n)} \right)^2$$

$$= \sum_{i=m_\ell+1}^{n} \left( \left\langle \widehat{\phi}_i, \sum_{j=1}^{m_\ell} \gamma_j \phi_j \right\rangle_{\rho(n)} + \left\langle \widehat{\phi}_i, f^* - \sum_{j=1}^{m_\ell} \gamma_j \phi_j \right\rangle_{\rho(n)} \right)^2$$

$$\leq 2 \sum_{i=m_\ell+1}^{n} \left( \left\langle \widehat{\phi}_i, \sum_{j=1}^{m_\ell} \gamma_j \phi_j \right\rangle_{\rho(n)} \right)^2 + 2 \sum_{i=m_\ell+1}^{n} \left( \left\langle \widehat{\phi}_i, f^* - \sum_{j=1}^{m_\ell} \gamma_j \phi_j \right\rangle_{\rho(n)} \right)^2, \tag{42}$$

where inequality (a) holds because that $0 < \widehat{\lambda}_i \leq 1$. The first term in (42) can be bounded as

$$\sum_{i=m_\ell+1}^{n} \left\langle \widehat{\phi}_i, \sum_{j=1}^{m_\ell} \gamma_j \phi_j \right\rangle_{\rho(n)}^2 \overset{(a)}{\leq} \sum_{i=m_\ell+1}^{n} \left( \sum_{j=1}^{m_\ell} \gamma_j^2 \right) \sum_{j=1}^{m_\ell} \left\langle \widehat{\phi}_i, \phi_j \right\rangle_{\rho(n)}^2 \overset{(b)}{\leq} \sum_{i=m_\ell+1}^{n} \sum_{j=1}^{m_\ell} \left\langle \widehat{\phi}_i, \phi_j \right\rangle_{\rho(n)}^2, \tag{43}$$

where inequality (a) follows from Cauchy-Schwarz inequality, and inequality (b) is true because that $\sum_{j=1}^{m_\ell} \gamma_j^2 \leq 1$. In addition, for any $\delta \in (0, 1)$, with probability at least $1 - \frac{\delta}{2}$, it holds that

$$\sum_{i=m_\ell+1}^{n} \sum_{j=1}^{m_\ell} \left\langle \widehat{\phi}_i, \phi_j \right\rangle_{\rho(n)}^2 \leq \frac{64 \widehat{\lambda}_{m_\ell+1} \log \frac{2}{\delta}}{\lambda_{m_\ell} (\lambda_{m_\ell} - \lambda_{m_\ell+1})^2 n}. \tag{44}$$

We postpone the proof of (44) to Section F.1. We bound the second term in (42) as

$$\sum_{i=m_\ell+1}^{n} \left\langle \widehat{\phi}_i, f^* - \sum_{j=1}^{m_\ell} \gamma_j \phi_j \right\rangle_{\rho(n)}^2 \leq \frac{1}{n} \sum_{k=1}^{n} \left( f^*(x_k) - \sum_{j=1}^{m_\ell} \gamma_j \phi_j(x_k) \right)^2 \leq \epsilon^2(f^*, \ell). \tag{45}$$

In addition, by (23) and the assumption that $n > \frac{256 \log \frac{2}{\delta}}{(\lambda_{m_\ell} - \lambda_{m_\ell+1})^2}$, with probability at least $1 - \delta$,

$$\widehat{\lambda}_{m_\ell} \geq \frac{3}{4} \lambda_{m_\ell}, \text{ and } \widehat{\lambda}_{m_\ell+1} \leq \lambda_{m_\ell}. \tag{46}$$

By (42), (43), (44), (45), and (46), we continue to bound (41) as: for any $\delta \in (0, \frac{1}{4})$, with probability at least $1 - 2\delta$,

$$\left\| \frac{1}{\sqrt{n}} \left( \boldsymbol{I} - \eta \boldsymbol{K} \right)^t \boldsymbol{y} \right\|^2 \leq \left( 1 - \frac{3\eta}{4} \lambda_{m_\ell} \right)^{2t} + \frac{128 \log 2/\delta}{(\lambda_{m_\ell} - \lambda_{m_\ell+1})^2 n} + 2\epsilon^2(\ell, f^*).$$

### F.1 Proof of Eq. (44)

**Preliminaries** Recall from (23) that the spectrum of $\boldsymbol{K}$ concentrates on the spectrum of the integral operator $L_{\mathcal{K}}$. To show (44), we need to know how $\phi_i$, $i \geq 1$ the eigenfunctions of $L_{\mathcal{K}}$ and $\widehat{\phi}_i$, $1 \leq i \leq n$ the eigenfunctions of $\boldsymbol{K}$ are related. Though both $L_{\mathcal{K}}$ and $\boldsymbol{K}$ are defined w. r. t. the kernel function $\mathcal{K}$ (defined in (12)), investigating this relation is not easy. This is because that $\phi_i$ is defined on $L^2(\mathcal{S}^{d-1}, \rho)$, whereas $\widehat{\phi}_i$ is defined on $L^2(\mathcal{S}^{d-1}, \rho(n))$. To overcome this difficulty, we relate $L_{\mathcal{K}}$ and $\boldsymbol{K}$ to two linear operators $T_{\mathcal{H}}$ and $T_n$, respectively, on $\mathcal{H}$ the *reproducing kernel Hilbert space* (RKHS) associated with the kernel function $\mathcal{K}$. In particular, we define $T_{\mathcal{H}}$ and $T_n$ by

$$T_{\mathcal{H}}f = \int_{\mathcal{S}^{d-1}} \langle f, \mathcal{K}_x \rangle_{\mathcal{H}} \mathcal{K}_x \mathrm{d}\rho(x), \text{ and } T_n f = \frac{1}{n} \sum_{i=1}^n \langle \cdot, \mathcal{K}_{x_i} \rangle_{\mathcal{H}} \mathcal{K}_{x_i}.$$

Here $\langle \cdot, \cdot \rangle_{\mathcal{H}}$ is the inner product with the RKHS $\mathcal{H}$ that satisfies $f(x) = \langle f, \mathcal{K}_x \rangle_{\mathcal{H}}$ for $f \in \mathcal{H}$, where $\mathcal{K}_x = \mathcal{K}(x, \cdot)$. It has been shown that the spectra of $L_{\mathcal{K}}$ and $T_{\mathcal{H}}$ are the same, possibly up to the zero, and that the spectra of $\boldsymbol{K}$ and $T_n$ are the same, possibly up to the zero. More importantly, clear correspondences between $L_{\mathcal{K}}$ and $T_{\mathcal{H}}$ and between $\boldsymbol{K}$ and $T_n$ are established. See [RBV10, item 2 of Proposition 8] and [RBV10, item 2 of Proposition 9] for details. Notably, there is a notational issue in [RBV10] which leads to an error in the multipliers in the correspondences. But this error can be fixed easily, and our calculation reflects this correction.

**Proof** We first show that $\sum_{i=m_\ell+1}^n \sum_{j=1}^{m_\ell} \left\langle \widehat{\phi}_i, \phi_j \right\rangle_{\rho(n)}^2$ can be upper bounded with (1) the difference between the projection of $T_{\mathcal{H}}$ onto its first $m_\ell$ eigenfunctions and that of $T_n$, and (2) the correspondences between the eigenfunctions of $L_{\mathcal{K}}$ and $T_{\mathcal{H}}$ and between that of $\boldsymbol{K}$ and $T_n$. Then we apply existing bound on the projection difference to conclude the proof.

Let $v_1, \cdots, v_{m_\ell}, \cdots$ be the orthonormal set of functions in $\mathcal{H}$ that related to $\phi_1, \cdots, \phi_{m_\ell}, \cdots$ by the relation given by [RBV10, item 2 of Proposition 8]. Similarly, let $\widehat{v}_1, \cdots, \widehat{v}_n$ be the corresponding Nystrom extension given by [RBV10, item 2 of Proposition 9]. Complete $\{v_i\}_{i \geq 1}$ and $\{\widehat{v}_i\}_{1 \leq i \leq n}$, respectively, to orthonormal bases of $\mathcal{H}$. Define two projection operators as follows:

$$P^{T_{\mathcal{H}}} = \sum_{j=1}^{m_\ell} \langle \cdot, v_j \rangle_{\mathcal{H}} v_j, \quad P^{T_n} = \sum_{j=1}^{m_\ell} \langle \cdot, \widehat{v}_j \rangle_{\mathcal{H}} \widehat{v}_j.$$

Since both $(v_j)_{j \geq 1}$ and $(\widehat{v}_j)_{j \geq 1}$ are orthonormal bases for $\mathcal{H}$, it is true that

$$\| P^{T_n} - P^{T_{\mathcal{H}}} \|_{HS}^2 = \sum_{i \geq 1, j \geq 1} \left| \left\langle \left( P^{T_n} - P^{T_{\mathcal{H}}} \right) \widehat{v}_i, v_j \right\rangle \right|^2,$$

where $\|\cdot\|_{HS}$ denotes the Hilbert–Schmidt norm defined as $\|A\|_{HS}^2 = \sum_{i \in I} \|A e_i\|^2$ for an orthonormal basis $\{e_i : i \in I\}$. By definition of $P^{T_{\mathcal{H}}}$ and $P^{T_n}$, we have

$$\left\langle \left( P^{T_n} - P^{T_{\mathcal{H}}} \right) \widehat{v}_i, v_j \right\rangle = \left\langle P^{T_n} \widehat{v}_i, v_j \right\rangle - \left\langle P^{T_{\mathcal{H}}} \widehat{v}_i, v_j \right\rangle = \begin{cases} 0, & \text{if } 1 \leq i \leq m_\ell, \& 1 \leq j \leq m_\ell; \\ \langle \widehat{v}_i, v_j \rangle_{\mathcal{H}}, & \text{if } 1 \leq i \leq m_\ell, \& j \geq m_\ell + 1; \\ -\langle \widehat{v}_i, v_j \rangle_{\mathcal{H}}, & \text{if } i \geq m_\ell + 1, \& 1 \leq j \leq m_\ell; \\ 0, & \text{if } i \geq m_\ell + 1, \& j \geq m_\ell + 1. \end{cases}$$

Thus we get

$$\| P^{T_n} - P^{T_{\mathcal{H}}} \|_{HS}^2 = \sum_{i=1}^{m_\ell} \sum_{j \geq m_\ell + 1} \left( \langle \widehat{v}_i, v_j \rangle_{\mathcal{H}} \right)^2 + \sum_{i \geq m_\ell + 1} \sum_{j=1}^{m_\ell} \left( \langle \widehat{v}_i, v_j \rangle_{\mathcal{H}} \right)^2 \geq \sum_{i=m_\ell+1}^n \sum_{j=1}^{m_\ell} \left( \langle \widehat{v}_i, v_j \rangle_{\mathcal{H}} \right)^2.$$

Since with probability 1 over the data generation $\widehat{\lambda}_i > 0$ for $i = 1, \cdots, n$, for $1 \leq i \leq n$, we have

$$\left( \langle \widehat{v}_i, v_j \rangle_{\mathcal{H}} \right)^2 = \frac{1}{\widehat{\lambda}_i} \left\langle \widehat{\phi}_i, v_j \right\rangle_{\rho(n)}^2.$$

So it holds that

$$\sum_{i=m_\ell+1}^n \sum_{j=1}^{m_\ell} \left(\langle \widehat{v}_i, v_j\rangle_{\mathcal{H}}\right)^2 \geq \frac{1}{\widehat{\lambda}_{m_\ell+1}} \sum_{i=m_\ell+1}^n \sum_{j=1}^{m_\ell} \left(\left\langle \widehat{\phi}_i, v_j\right\rangle_{\rho(n)}\right)^2 \overset{(b)}{=} \frac{1}{\widehat{\lambda}_{m_\ell+1}} \sum_{i=m_\ell+1}^n \sum_{j=1}^{m_\ell} \left(\left\langle \widehat{\phi}_i, \sqrt{\lambda_j}\phi_j\right\rangle_{\rho(n)}\right)^2$$

$$\geq \frac{\lambda_{m_\ell}}{\widehat{\lambda}_{m_\ell+1}} \sum_{i=m_\ell+1}^n \sum_{j=1}^{m_\ell} \left(\left\langle \widehat{\phi}_i, \phi_j\right\rangle_{\rho(n)}\right)^2,$$

where equality (b) follows from [RBV10, Proposition 8, item 2]. Since $n > \frac{256 \log \frac{2}{\delta}}{(\lambda_{m_\ell} - \lambda_{m_\ell+1})^2}$, by [RBV10, Theorem 7 and Proposition 6], it holds that with probability at least $1 - \frac{\delta}{2}$,

$$\|P^{T_n} - P^{T_{\mathcal{H}}}\|_{HS}^2 \leq \frac{64 \log \frac{2}{\delta}}{(\lambda_{m_\ell} - \lambda_{m_\ell+1})^2 n},$$

finishing the proof of Eq. (44).

$\square$

# G   Harmonic analysis on spheres

Throughout this section, we consider uniform distribution $\rho$ on the unit sphere in $\mathbb{R}^d$ with $d \geq 3$, and we consider functions on on $\mathcal{S}^{d-1}$. For ease of exposition, we do not explicitly write out the dependence on $d$ in the notations.

Let $\mathcal{H}_\ell$ denote the space of degree-$\ell$ homogeneous harmonic polynomials on $\mathcal{S}^{d-1}$:

$$\mathcal{H}_\ell = \left\{ P : \mathcal{S}^{d-1} \mapsto \mathbb{R} : P(x) = \sum_{|\alpha|=\ell} c_\alpha x^\alpha, \Delta P = 0 \right\},$$

where $x^\alpha = x_1^{\alpha_1} \cdots x_d^{\alpha_d}$ is a monomial with degree $|\alpha| = \alpha_1 + \cdots + \alpha_d$, $c_\alpha \in \mathbb{R}$, and $\Delta$ is the Laplacian operator. The dimension of $\mathcal{H}_\ell$ is denoted by $N_\ell = \frac{(2\ell+d-2)(\ell+d-3)!}{\ell!(d-2)!}$. For any $\ell$ and $\ell'$, the spaces $\mathcal{H}_\ell$ and $\mathcal{H}_{\ell'}$ are orthogonal to each other.

The Gegenbauer polynomials, denoted by $C_\ell^{(\lambda)}$ for $\lambda > -\frac{1}{2}$ and $\ell = 0, 1, \cdots$, are defined on $[-1, 1]$ as

$$C_\ell^{(\lambda)}(u) = \sum_{k=0}^{\lfloor \ell/2 \rfloor} (-1)^k \frac{\Gamma(\ell - k + \lambda)}{\Gamma(\lambda)k!(\ell - 2k)!}(2u)^{\ell-2k}, \tag{47}$$

where $\Gamma(v) := \int_0^\infty z^{v-1}e^{-z}\mathrm{d}z$. Notably, $\Gamma(v + 1) = z\Gamma(v)$. The cases $\lambda = 0, \frac{1}{2}, 1$ correspond to Chebyshev polynomials of the first kind, Legendre polynomials, Chebyshev polynomials of the second kind, respectively. It has been shown that [Sze75, Section 4.1(2), Section 4.7] for $\lambda \neq 0$ Gegenbauer polynomials are orthogonal with the weight function $w_\lambda(u) = (1 - u^2)^{\lambda - \frac{1}{2}}$:

$$\int_{-1}^1 C_\ell^{(\lambda)}(u)C_k^{(\lambda)}(u)w_\lambda(u)\mathrm{d}u = \frac{\pi 2^{1-2\lambda}\Gamma(\ell + 2\lambda)}{\ell!(\ell + \lambda)(\Gamma(\lambda))^2}\delta_{k,\ell},$$

where $\delta_{k,\ell} = 1$ if $k = \ell$ and $\delta_{k,\ell} = 0$ otherwise. The orthogonality can be equivalently written as

$$\int_0^\pi C_\ell^{(\lambda)}(\cos\theta)C_k^{(\lambda)}(\cos\theta)\sin^{2\lambda}\theta\mathrm{d}\theta = \frac{\pi 2^{1-2\lambda}\Gamma(\ell + 2\lambda)}{\ell!(\ell + \lambda)(\Gamma(\lambda))^2}\delta_{k,\ell}.$$

For each $\ell \in \mathbb{N}$, there exists a set of orthonormal basis $\{Y_{\ell,i} : i = 1, \ldots, N_\ell\}$ for $\mathcal{H}_\ell$ w.r.t. the uniform distribution $\rho$ that can be written in terms of $C_\ell^{(\lambda)}$ in [DX13, Theorem 1.5.1] as

$$C_\ell^{(\lambda)}(\langle x, y\rangle) = \frac{\lambda}{\ell + \lambda} \sum_{i=1}^{N_\ell} Y_{\ell,i}(x)Y_{\ell,i}(y), \quad \lambda = \frac{d - 2}{2}. \tag{48}$$

This is known as the *addition theorem*. Therefore, a function of the form $f(x, y) = f(\langle x, y \rangle)$ (i.e., the value of $f(x, y)$ depends on $x$ and $y$ through their angle $\langle x, y \rangle$ only) can be expanded under $C_\ell^{(\lambda)}$ as

$$f(x, y) = f(\langle x, y \rangle) = \sum_{\ell \geq 0} \alpha_\ell C_\ell^{(\lambda)}(u) = \sum_{\ell \geq 0} \frac{\alpha_\ell \lambda}{\ell + \lambda} \sum_{i=1}^{N_\ell} Y_{\ell,i}(x) Y_{\ell,i}(y) , \qquad (49)$$

where $u = \langle x, y \rangle$, $\lambda = \frac{d-2}{2}$, and

$$\alpha_\ell = \frac{\int_{-1}^{1} f(u) C_\ell^{(\lambda)}(u) w_\lambda(u) \mathrm{d}u}{\int_{-1}^{1} (C_\ell^{(\lambda)}(u))^2 w_\lambda(u) \mathrm{d}u}.$$

For the kernel function defined in (16), it can be expanded as

$$\mathcal{K}(x, s) = \sum_{\ell \geq 0} \beta_\ell \sum_{i=1}^{N_\ell} Y_{\ell,i}(x) Y_{\ell,i}(s), \quad \text{where } \beta_\ell := \frac{\alpha_\ell \frac{d-2}{2}}{\ell + \frac{d-2}{2}},$$

where for each $\ell \geq 0$, $\alpha_\ell$ is the coefficient of $\mathcal{K}(x, s)$ in the expansion into Gegenbauer polynomials, $\beta_\ell$ is the eigenvalue associated with the space of degree–$\ell$ homogeneous harmonic polynomials on $\mathcal{S}^{d-1}$, denoted by $\mathcal{H}^\ell$, and $Y_{\ell,i}$ for $i = 1, \cdots, N_\ell$ are an orthonormal basis of $\mathcal{H}^\ell$. Thus, the corresponding integral operator can be decomposed as $L_\mathcal{K} = \sum_{\ell \geq 0} \beta_\ell P_\ell$.

## Acknowledgement

We would like to thank Yang Yuan (Tsinghua IIIS) for his insightful initial discussions, and Rong Ge (Duke) for suggesting the network initialization rule.