[Reviews · NeurIPS 2019]

Reviewer 1



Recent theoretical work on analyzing Deep Neural Network training has focused on the evolutions of the over-parametrized neural network prediction errors under Gradient Descent. These evolutions can be neatly described in a matrix form. This paper found that the matrices can approximate an integral operator this is determined by the feature vector distribution only. Consequently, Gradient Method can be viewed as approximately apply the powers of this integral operator on the underlying function that generates the labels. Moreover, this paper also derives a new linear convergence rate on the new assumption of the underlying function. This analysis is derived from the integral operator approximation analysis above. This paper is well-organized and well-written. The novelty, clarity, and originality of this paper are quite great. This paper provides a new tool to understand why the basic Gradient Descent methods work very well on Deep Learing models and showed new convergence rates on the new assumptions of the underlying function and feature vector generation, which might be fit to situations in the real world.

Reviewer 2



One of the biggest reasons that I am not too thrilled about the submission is that the two-layer fully connected neural network model under consideration is off standard; the weights of the second layer is fixed to binary (i.e. +1, -1 with uniform scaling) and are not being updated via the gradient descent procedures. Correct me if I am wrong; this assumption is neither commonly adopted in experiments nor identical to the comparable theoretical works such as [DLL+18] or [ADH+19]. If the authors are convinced of the significance or general applicability of the suggested framework, they should have taken more care communicating those to the audience. A relatively minor issue is about the significance of the c_1 term in Theorem 3. The authors nicely demonstrate that the constant c_1 satisfying (13) can be controlled via a sum whose dominating term is inversely proportional to (lambda_{m_l} - lambda_{m_l + 1}), which is later formalized to Theorem 4. (By the way, I had trouble locating a formal definition of eps(f^*,l). ) The question is, can we guarantee that the value is large enough to ensure the ignorability of the term c_1? I am particularly worried about this, as the authors have already mentioned that the spectrum of the random matrix concentrates as n grows, in line 145. Beside these issues, technical clarity and validity of the submission seems satisfactory, and the proof ideas would be of at least mild interest to the theoreticians concerned with deep learning. Plus, the authors do a nice job of criticizing the previous results. +) please do the grammar check, to remove the typos including: line 7, 'approximately apply'; line 32, 'prediction errors evolution'; line 32, 'neuronscience'; line 48, 'approximately apply' (stopped counting around here...).

Reviewer 3



This paper considers gradient descent as an operator on functions, and thereby obtains convergence guarantees in terms of the eigenvalues of that operator. A similar perspective, in a somewhat simpler setting, was studied in Vempala--Wilmes (2017). Convergence guarantees for empirical risk minimization where also obtained in the overparameterized setting in Arora et. al. (2019), among others. The authors observe that some previous convergence guarantees give rates that tend to zero as the number of examples increases. To avoid this problem, the authors introduce the assumption that the target function is well approximated by its projection to a small number of eigenspaces of the gradient operator. For example, the function could be a harmonic polynomial over the sphere with respect to the uniform distribution. In this case, the authors obtain linear convergence rate as long as the number of hidden units is at least quadratic in the number of examples. Overall, this is a pleasant paper, mathematically. However, its improvements over the existing literature are not dramatic. --- I thank the authors for their very careful response. I continue to believe this paper is worth accepting to NeurIPS and have slightly improved my score.

[Author Response · NeurIPS 2019]

**Summary** We would like to thank the entire review team for their efforts and insightful comments. In particular, we would like to thank Reviewer 1 for the positive comments, and Reviewers 2 and 3 for sharing their concerns on the significance of our results compared with existing literature. We sincerely apologize for the lack of clarity of our original submission in distinguishing our results from existing work. We will take more care of communicating the improvements in a revised version if we get a chance.

Overall, the advantages of taking such a functional approximation perspective are at least three-fold: It can
(A) help us spot the fact that some existing convergence guarantees are diminishing in the sample size $n$,
(B) single out the impacts of the properties of the true function $f^*$ on the convergence speed, and
(C) improve the state-of-the-art results on the sufficiency of network over-parameterization.
Below, we first detail the significance of these three advantages, and then provide clarifications on the specific issues raised by the reviewers.

**Advantages of adopting a functional approximation perspective**

**(A):** We showed in Theorem 2 that the existing rate characterizations in the influential line of work [ADH+19, DLL+18, DZPS18] ([DZPS18] refers to arXiv:1810.02054) approach zero (i.e., $\to 0$) as the sample size $n \to \infty$. In fact, even the rate derived in the state-of-the-art work on training over-parameterized neural networks (NNs) [OS19] approaches zero as $n \to \infty$; see Corollary 2.2. in [OS19]. However, in many applications the volumes of the datasets are huge – the ImageNet dataset has 14 million images. For those applications, a non-diminishing convergence rate is more desirable.

**(B):** Recall that $f^*$ denotes the underlying function that generates output labels/responses (i.e., $y$'s) given input features (i.e., $x$'s). For example, $f^*$ could be a constant function or a linear function, i.e., $f^*(x) \equiv c$ or $f^*(x) = \theta^\top x$. Clearly, the difficulty in learning $f^*$ via training neural networks should crucially depend on the properties of $f^*$ itself. Our Theorem 4 and Corollary 2 essentially say that the training convergence rate is determined by how $f^*$ can be decomposed into the eigenspaces of some integral operator. Our results are also validated by a couple of existing empirical observations: (1) The spectrum of the MNIST data concentrates on the first a few eigenspaces; and (2) the training is slowed down if labels are partially corrupted [Zhang et al. 2016] (arXiv:1611.03530). One important practical implication of our results is: in order to speed up training, the practitioners could "adapt" the eigenspaces of the underlying integral operator of GD by designing better feature engineering method so that the underlying true function could be well projected onto a few eigenspaces.

**(C):** It has been empirically observed that linear over-parameterization $m = \Theta(n)$ is sufficient for GD to converge [ZBHB16]. However, the state-of-the-art theoretical results on network over-parameterization is $m = \Theta(n^2)$ but at a price of having diminishing convergence rate (Corollary 2.2. in [OS19]). In our work, we show (in Corollary 2) that if $f^*$ can be decomposed into a finite number of eigenspaces of the integral operator, then $m = \Theta(n^2)$ is sufficient and a constant convergence rate can be achieved. Moreover, we conjecture (not mentioned in our original submission) that with a slightly different network initialization, the over-parameterization level might be improved to $m = \Theta(n\,\mathsf{poly}(\log n))$. In particular, for each hidden unit $j$ (where $j = 1, \cdots, m$), we introduce a pairing hidden unit $j'$. We initialize $w_j$ and $a_j$ as there were in our original submission, and set $w_{j'}^0 = w_j^0$ and $a_{j'}^0 = -a_j^0$ for each $j$. By Eq. (4), we know $\hat{y}_i(0) = 0$ for $i = 1, \cdots, m$; thus, we do not need to set $\nu$ to be small in order to control $\|\hat{y}_i(0)\|$. Besides, only the first three terms in the upper bound of $\|\frac{1}{\sqrt{n}}(\boldsymbol{I} - \eta\boldsymbol{K})^t\boldsymbol{y}\|$ in Lemma 5 remain.

**Response to the concern on fixed second layer.** We would like to thank Reviewer 2 for raising this question, and we sincerely apologize for the lack of justification in our original submission. This assumption is indeed frequently used in many theoretical works. Specifically, the same assumption is made in [ADH+19] and [ZCZG18] (arXiv:1811.08888), the later of which studied the general deep nets. Similar frozen assumption is adopted in [ALS18] (arXiv:1811.03962). We do agree this assumption might restrict the applicability of our results. Nevertheless, even this setting is not well-understood despite the recent intensive efforts. Our analysis might be generalizable to the setting wherein both layers are jointly optimized, and the output layer is initialized by Glorot/He initialization: For the general setting, the kernel of the integral operator is a sum of two kernel component functions – the additional kernel component function captures the mutual "interruption" of different weights at the second layer under GD method. Since both of the two kernels are positive semidefinite, we can bound the second kernel function by zero mapping. Then we can follow the line of analysis in the current paper to conclude.

**Response to the significance of $c_1$.** Sorry for the confusion. We will emphasize the definition of $\epsilon(f^*, \ell)$, and the notation $\lambda_{m_\ell}$ and $\lambda_{m_\ell+1}$ in a revised version if possible. Here $\lambda_{m_\ell}$ and $\lambda_{m_\ell+1}$ (introduced in the paragraph above Theorem 4) are the $\ell$–th and $\ell + 1$–th largest *distinct* eigenvalues of the integral operator, and $(\lambda_{m_\ell} - \lambda_{m_\ell+1})$ is the eigengap. Once the distribution $\rho$ is fixed, the eigenvalues of the integral operator is also fixed – they do not change with the sample size $n$. Thus, for fixed $\rho$ and $f^*$, $c_1 = \Theta(\sqrt{\log(1/\delta)/n})$.

**Response to other issues.** Due to space limit we collectively respond to other issues here. We fixed grammars and typos mentioned by Reviewer 2, and carefully went through the entire article to fix others in a revised version; we also clarified notations at their first appearances as pointed out by Reviewer 3.

[Meta-Review · NeurIPS 2019]

This paper gives convergence guarantees for training neural networks via gradient descent. The approach consists of considering GD as an operator and of analyzing it through its eigenvalues. The interest of the analysis is focus on the overparameterized setting of such network. This is an interesting paper, with interesting theoretical results. It is clearly above the acceptance threshold. We are not recommending any oral presentation, because, as it is the paper proposed improvements are not important enough when comparing with the existing literature. Namely, in the discussion phase, the reviewers agreed that the authors should cite the papers recommended by Reviewer #3: “A similar perspective, in a somewhat simpler setting, was studied in Vempala--Wilmes (2017). Convergence guarantees for empirical risk minimization where also obtained in the overparameterized setting in Arora et. al. (2019), among others.”